# Mind the State: Towards Unified, Context-Aware EEG-to-fMRI Synthesis

**Yamin Li** [1]  **Shiyu Wang** [1]  **Chang Li** [1]  **Ange Lou** [1]  **Haatef Pourmotabbed** [1]  **Sarah Goodale** [1,2]  **Dario J. Englot** [1,2]
**Daniel Moyer** [1]  **Roza G. Bayrak** [1]  **Catie Chang** [1]

## Abstract

Functional magnetic resonance imaging (fMRI) provides dynamic measurements of human brain activity at high spatial resolution and depth, but its use is constrained by high cost, limited accessibility, and strict acquisition requirements. Synthesizing fMRI data from more accessible, non-invasive modalities such as electroencephalography (EEG) offers a promising alternative, enabling inference of deep brain dynamics from low-cost scalp recordings in naturalistic settings. Despite recent progress, existing EEG-to-fMRI translation methods typically rely on region-specific models and offer limited support for subject-level and dataset-level heterogeneity, restricting their generalizability. We propose **UniEFS**, a **uni**fied **E**EG-to-**f**MRI **S**ynthesis model that enables full-brain fMRI reconstruction while accommodating varying demographic and physiological contexts within a single model. Our approach leverages a pretrained fMRI decoder to embed rich spatial priors and introduces condition-aware prompt tokens that encode subject-level and experimental metadata, enabling effective handling of heterogeneous datasets. We extensively evaluate the model performance on eyes-closed resting-state data and demonstrate that it can reliably reconstruct temporally-resolved whole-brain fMRI activity, with potential to generalize to task-based fMRI and clinical populations in a zero-shot manner. Project page: https://soupeeli.github.io/UniEFS

## 1. Introduction

The ability to non-invasively monitor brain activity is essential for advancing both neuroscience research and clinical care. Electroencephalography (EEG) and functional magnetic resonance imaging (fMRI) represent two ends of the neuroimaging spectrum. EEG captures fast, millisecond-scale electrical signals from the scalp, offering a direct window into neural activity with excellent temporal resolution and broad accessibility (Nicolas-Alonso & Gomez-Gil, 2012; Tong & Thakor, 2009; Li et al., 2020). However, it suffers from poor spatial resolution and limited sensitivity for mapping large-scale and deep-brain circuits (Cohen, 2017; Chang & Chen, 2021). In contrast, fMRI provides rich spatial detail by measuring blood oxygenation-level dependent (BOLD) signals driven by neurovascular coupling across the entire 3D volume of the brain (Logothetis, 2008; Matthews et al., 2006). Yet fMRI is expensive, infrastructure-intensive, and constrained by low temporal resolution. It is also largely inaccessible in under-resourced communities and outpatient settings, and may be contraindicated for patients with certain implants or conditions (Jalloul et al., 2023; van Beek et al., 2019; Geethanath & Vaughan Jr, 2019). These complementary characteristics raise an intriguing question: *Can we equip EEG with fMRI-like representational power?* If so, it would unlock a new paradigm for scalable, high-resolution brain monitoring and decoding using only a lightweight, real-time, and non-invasive sensor, transforming both clinical practice and cognitive neuroscience.

These factors motivate a growing interest in reconstructing fMRI signals from EEG, leveraging their underlying physiological correlation and the representational power of deep learning to bridge the spatial and temporal divide between these two modalities. A particularly underexplored area in this field involves the *eyes-closed, resting-state condition*. This condition is of significant interest in both research and clinical contexts: it offers a window into the brain's intrinsic functional organization and is widely used due to its simplicity and ease of implementation. It is especially valuable for its applicability to diverse populations, including children and patients who may not tolerate or comply with task-based paradigms. However, unlike task-based paradigms that provide clear temporal anchors, resting-state brain activity is more spontaneous and variable, spanning a variety of internal brain states such as mind-wandering, vigilance fluctuations, or even light sleep, which makes it inherently more challenging to decode (Liu, 2016; Liu &

[1]Vanderbilt University, TN, USA [2]Vanderbilt University Medical Center, TN, USA. Correspondence to: Yamin Li <yamin.li@vanderbilt.edu>, Catie Chang <catie.chang@vanderbilt.edu>.

*Proceedings of the 43rd International Conference on Machine Learning*, Seoul, South Korea. PMLR 306, 2026. Copyright 2026 by the author(s).

Falahpour, 2020). For example, changes in vigilance introduce substantial non-stationarity: the decline from alertness into drowsiness and light sleep is accompanied by marked changes in the spectral content of EEG, and in the signal amplitude and network structure of fMRI (Liu & Falahpour, 2020; Martin et al., 2021; Li et al., 2025).

Within the body of existing work, NeuroBOLT (Li et al., 2024b) is, to our knowledge, the only published study to date that has explored EEG-fMRI translation under the eyes-closed resting-state condition. While demonstrating promising results with multi-dimensional EEG representation learning, it requires training separate models for individual brain regions, limiting efficiency and scalability. Another very recent approach, CATD (Yao et al., 2025), demonstrated efficient cortical surface fMRI generation by conditioning a diffusion model on EEG. However, by design, its reliance on fMRI surface maps restricts reconstruction to the cortex, leaving subcortical regions outside the model's representational space. Subcortical brain areas are increasingly recognized as vital to healthy cognition as well as a wide range of disease processes (Favaretto et al., 2022; Koshiyama et al., 2018; Shepherd, 2013). This work also included resting-state data, albeit during eyes-open conditions, which may not be as conducive to more dramatic shifts in vigilance (e.g., falling asleep) compared to eyes-closed conditions. Moreover, these approaches share a common limitation: they treat the EEG-fMRI relationship as largely uniform across individuals. This overlooks inter-subject variability driven by demographic factors (e.g., age and sex) and by dynamic, time-varying physiological states (e.g., drowsiness or vigilance), which are particularly pronounced during resting-state recordings. Such states have been shown to modulate both EEG and fMRI signals, as well as their correlations (Liu & Falahpour, 2020; Olbrich et al., 2009; Wong et al., 2013; Wang et al., 2025b). Addressing this variability is therefore essential for developing scalable and generalizable EEG-to-fMRI translation frameworks that extend beyond specific conditions or cohorts. Yet existing methods have been evaluated primarily within their training domains, leaving generalizability largely unexplored. A more comprehensive review of related work is provided in Appendix A.

To address these challenges, we propose a unified, context-aware framework for generalizable and efficient EEG-to-fMRI translation. Rather than following prior work that relies solely on end-to-end training with scarce paired EEG-fMRI data, we adopt a two-stage strategy: **(1) uni-modal fMRI pretraining**, and **(2) cross-modal alignment**. **Stage (1)** focuses on learning expressive and population-level fMRI representations from larger-scale unpaired fMRI data with rich coverage of brain dynamics. A key motivation stems from the observation that fMRI activity exhibits structured spatial patterns even at the level of individual frames

(Liu et al., 2018; 2013). In particular, co-activation pattern (CAP) analyses have revealed that groups of brain regions display recurring and instantaneous configurations of activation and deactivation (Liu et al., 2018). Building on this insight, we design a self-supervised masked modeling strategy (Chen et al., 2023; Xie et al., 2022; Radford et al., 2019; Yang et al., 2023; Jiang et al., 2024) that trains the model to recover masked brain regions from the visible context within each frame, encouraging the model to capture transient spatial dependencies across regions and learn robust representations of instantaneous brain states. In **Stage (2)**, we align EEG with this learned fMRI latent space and reconstruct fMRI with the pretrained decoder. To facilitate this, we introduce a context-aware EEG encoder that projects temporal and spectral features into the pretrained fMRI space, while explicitly incorporating auxiliary metadata. This contextual conditioning enables the model to account for individual variability in the EEG-fMRI relationship, thereby bridging the two modalities in a unified framework. UniEFS enables full-brain fMRI reconstruction without region-specific supervision or subject-dependent customization, offering an effective and scalable framework for decoding intrinsic brain activity under eyes-closed resting-state conditions and beyond. The key contributions are summarized as follows:

**Whole-brain EEG-to-fMRI synthesis.** We develop a unified framework that reconstructs fMRI activity, spanning hundreds of functional regions, from EEG using a single model. By first pretraining the fMRI decoder on unpaired fMRI data, we embed strong spatial priors to promote accurate and efficient reconstruction across brain regions.

**Context-aware EEG encoding.** To better accommodate heterogeneity in EEG-fMRI data (e.g., different demographic attributes, and physiological states), we introduce prefix prompt tokens that encode dataset-specific and subject-level metadata, facilitating unified training across formats.

**Systematic evaluation of the predictive power.** We conduct extensive evaluations of fMRI time-series reconstruction performance across multiple brain areas, including cortical and subcortical regions, as well as whole-brain functional connectivity patterns. Beyond within-domain evaluations, we assess the model's ability to generalize across several experimental conditions, examining zero-shot transfer performance and predictive capacity.

## 2. Methods

### 2.1. Overview

In this section, we describe the overall task setting and proposed framework of UniEFS. Our approach performs frame-by-frame fMRI prediction: given a sliding window of EEG preceding each fMRI time point, the model predicts

the corresponding fMRI frame. This design enables flexible generation of fMRI sequences of arbitrary length. Our work mainly focuses on Regions-of-Interest-level (ROI-level) reconstruction, which offers a favorable trade-off between spatial resolution and efficiency. Compared to voxel-wise and surface-based methods, it reduces computational cost and improves signal-to-noise ratio (SNR), while also covering both cortical and subcortical regions for full-brain modeling. Moreover, as a representation adopted in recent fMRI foundation models (Dong et al., 2024; Caro et al., 2024; Thomas et al., 2022), ROI-level modeling provides a scalable and effective basis for future extensions.

However, achieving accurate frame-wise ROI-level reconstruction from EEG is non-trivial, due to the following key challenges. First, frame-wise reconstruction implicitly involves learning the projection from neuronal activity to its hemodynamic response, which varies across brain regions, individuals, and brain states. Second, paired EEG–fMRI datasets are scarce and moreover vary in their subject characteristics, potentially hindering generalization to broader populations and different conditions. To address these challenges, we propose a two-stage learning framework as illustrated in Figure 1: **(1) fMRI Pretraining and Adaptation via Masked Signal Modeling:** We first pretrain a powerful encoder-decoder model on unpaired fMRI datasets using a masked reconstruction objective. This stage enables the model to learn population-level representations of brain activity. To bridge the potential domain gap between pretraining and downstream application, we further fine-tune the pretrained model on the fMRI data from the training split of the paired EEG-fMRI dataset, adapting the decoder to the target domain while preserving its generalization capacity. **(2) Context-aware EEG-to-fMRI Mapping:** In the second stage, we integrate a dedicated EEG encoder, conditioned on demographic and physiological priors, with the adapted fMRI decoder. The EEG encoder learns to map temporal and spectral features of EEG signals into the corresponding fMRI latent space, enabling full-brain fMRI reconstruction.

## 2.2. Stage 1: fMRI Pretraining and Adaptation via Masked Signal Modeling (f-MSM)

We summarize the 4-dimensional fMRI data using the Dictionaries of Functional Modes (DiFuMo) parcellation (Dadi et al., 2020) with $P = 512$ ROIs, which provides fine-grained, whole-brain coverage. This yields an fMRI matrix denoted as $Y \in \mathbb{R}^{P \times K}$, where $K$ is the total number of time points. During pretraining, each 1D ROI vector corresponding to a single time point is treated as an individual training sample, yielding $K$ samples per fMRI scan. Each ROI vector consists of $P$ scalar regional activity values, which are projected into higher-dimensional ROI-level brain tokens through a 1D patch embedding layer. Positional embeddings are then added to encode regional identity. Although par-

cellation reduces voxel-level redundancy, functional dependencies and spatial correlations persist across brain regions due to the network-level organization of brain activity. To encourage the model to capture these intrinsic patterns, for each of the above ROI vectors, we adopt a high masking ratio (50%, see Appendix D.8 for a sensitivity analysis) during pretraining, forcing the network to infer random missing regional signals from the surrounding context. This design promotes the learning of expressive, population-level representations that generalize across individuals and tasks. For this reconstruction task, we employ a transformer-based architecture (Chen et al., 2023; 2024; Dosovitskiy et al., 2020). An encoder processes only the visible (unmasked) ROI tokens, while a lightweight decoder reconstructs the complete ROI vector based on the contextual information inferred from the unmasked regions. After pretraining , we adapt the model using the fMRI data from the training split of the paired dataset.

**Training Objective.** Following He et al. (2022), the reconstruction loss (MSE) is computed solely on the masked tokens for both pretraining and finetuning.

## 2.3. Stage 2: Context-aware EEG-to-fMRI Mapping

We extract the EEG window spanning a duration $T$, corresponding to the approximate latency of the hemodynamic response function (HRF), before each fMRI frame collection. This forms an EEG-fMRI paired input-output sample denoted as $\{X, Y^{(\text{paired})}\}$, where $X \in \mathbb{R}^{C \times T}$ represents the EEG input with $C$ channels and $T$ time points, and $Y^{(\text{paired})} \in \mathbb{R}^{P}$ denotes the corresponding parcellated fMRI ROI vector with $P$ ROIs. The EEG input $X$ is first processed by the EEG encoder $\mathcal{E}_{\text{EEG}}$ to generate a latent representation, which is then passed to the pretrained domain-adapted decoder $\mathcal{D}_{\text{fMRI}}$ obtained from f-MSM. Overall, given the full model $f_\theta(.)$, the overall fMRI reconstruction task can be formulated as $\hat{Y}_t^{(\text{paired})} = f_\theta(X_{t-T:t-1})$, where $\hat{Y}_t^{(\text{paired})} \in \mathbb{R}^{P}$ is the reconstructed fMRI frame at time index $t$.

**EEG Encoder.** Our objective is to enable EEG-driven fMRI reconstruction by aligning EEG representations with the fMRI latent embedding space. To achieve this, we adapt the multi-dimensional encoder from NeuroBOLT (Li et al., 2024b) as the backbone encoder, a transformer-based architecture designed to capture rich and complementary spatial, temporal, and multi-scale spectral information from EEG signals. We first segment an EEG window $X$ into non-overlapping patches using a window of length $w$, yielding a sequence of patches $x_{c,k} \in \mathbb{R}^{w}$ for each channel $c = 1, \ldots, C$ and patch index $k = 1, \ldots, \lfloor T/w \rfloor$. These patches are then fed into (i) spatiotemporal module (a pretrained EEG encoder adapted from the EEG foundation model LaBraM (Jiang et al., 2024)) and (ii) multi-scale spectral transformer modules to generate two EEG latent embed-

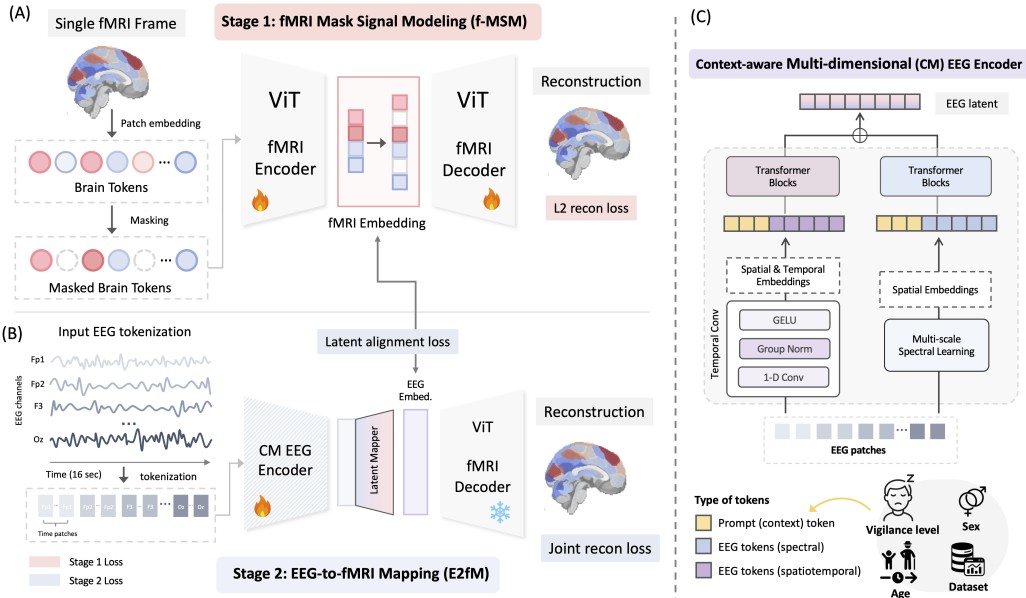

*Figure 1.* **Overall framework.** (A) Stage 1: Masked signal modeling on fMRI frames. (B) Stage 2: EEG-to-fMRI mapping. The pretrained fMRI encoder and decoder are frozen in this stage. (C) Context-aware EEG embedding.

dings $\mathbf{z}_{\mathrm{EEG}_{st}}, \mathbf{z}_{\mathrm{EEG}_{sp}} \in \mathbb{R}^{(C \times \frac{T}{w}) \times D}$ respectively, where $D$ is the embedding dimension. Instead of applying global average pooling across the token dimension (Li et al., 2024b; Jiang et al., 2024; Yang et al., 2023), we retain the full sequence of token embeddings to preserve fine-grained spatial and temporal information. The embeddings from two modules are summed up and then passed through a latent mapping module, which consists of two linear projections to align the EEG embeddings with the dimensionality and structure of the fMRI latent space. This final latent representation is then passed to the fine-tuned decoder $\mathcal{D}$ to reconstruct the full-brain fMRI signal.

**Prefix Prompt Injection.** To incorporate auxiliary information (Gao et al., 2024) and enhance the generalization of EEG representations, we introduce a set of learnable prefix prompts in the EEG encoder that are concatenated to the EEG patches prior to the transformer modules. These prompt tokens are designed to encode subject- and dataset-specific metadata and are optimized jointly with the rest of the model, enabling the network to adaptively condition its representation based on contextual information. Specifically, we include the following prompt tokens: **(1) Dataset tokens**: learnable embeddings of shape $\mathbb{R}^{J \times D}$, where $J$ is a tunable hyperparameter, representing the number of dataset tokens ($J = 5$ in our experiments). Each dataset is assigned its own set of tokens, enabling the model to capture dataset-specific characteristics. **(2) Age token**: A single token generated by projecting the subject's age through a linear embedding layer. **(3) Sex token**: A learnable token indicating the subject's biological sex (e.g., male or female). **(4) Vigilance token**: A categorical learnable token

encoding the vigilance level (drowsy, intermediate, alert) at each fMRI frame (TR), allowing the model to condition its reconstruction on frame-specific vigilance states. All prefix tokens share the same embedding dimension $D$ as the EEG patch embeddings output by the EEG encoder. Let $N_{\mathrm{prompt}}$ be the total number of prompt tokens used , such that $\mathbf{z}_{\mathrm{prompt}} \in \mathbb{R}^{N_{\mathrm{prompt}} \times D}$ represents all prompt tokens. These tokens are then concatenated to the EEG embedding $\mathbf{z}_{\mathrm{EEG}}$ along the token dimension to form the augmented token sequence $\mathbf{z} \in \mathbb{R}^{(N_{\mathrm{prompt}} + N_{\mathrm{EEG}}) \times D}$. This enriched EEG representation is then passed to the following transformers and to the alignment module for EEG-to-fMRI mapping.

**EEG-fMRI Embedding Alignment.** To align the EEG embeddings with the fMRI latent space, during training, we first obtain a reference embedding from fMRI by passing ground-truth fMRI vector $Y^{(\mathrm{paired})}$ through the frozen fine-tuned fMRI encoder $\mathcal{E}_{\mathrm{fMRI}}$. This yields the fMRI latent embedding:

$$\mathbf{E}_{\mathrm{fMRI}} = \mathcal{E}_{\mathrm{fMRI}}(Y^{(\mathrm{paired})}) \in \mathbb{R}^{P \times D_{\mathrm{fMRI}}}, \qquad (1)$$

where $D_{\mathrm{fMRI}}$ is the dimension of the fMRI latent space. In parallel, the EEG input $X$ is first processed by the EEG encoder $\mathcal{E}_{\mathrm{EEG}}$ to obtain latent embeddings. We then apply a linear projection $\mathcal{P}$ to map the enriched EEG embedding into the same shape as the fMRI embedding:

$$\mathbf{E}_{\mathrm{EEG}}^{\mathrm{proj}} = \mathcal{P}(\mathbf{E}_{\mathrm{EEG}}) \in \mathbb{R}^{P \times D_{\mathrm{fMRI}}}, \qquad (2)$$

which is then used in the alignment and decoding modules. To extract compact and semantically aligned representations between $\mathbf{E}_{\mathrm{fMRI}}$ and $\mathbf{E}_{\mathrm{EEG}}^{\mathrm{proj}}$, we adopt a LoRA-inspired

low-rank projection mechanism (Hu et al., 2022). Specifically, we define two learnable matrices: $\mathbf{B} \in \mathbb{R}^{D_{\text{fMRI}} \times r}$ and $\mathbf{A} \in \mathbb{R}^{r \times D_{\text{comp}}}$, where $r$ is the intermediate rank and $D_{\text{comp}}$ is the final compressed dimension. The final low-rank embeddings are computed as: $\tilde{\mathbf{E}}_{\text{fMRI}} = \mathbf{E}_{\text{fMRI}} \cdot \mathbf{B} \cdot \mathbf{A} \in \mathbb{R}^{P \times D_{\text{comp}}}$, $\tilde{\mathbf{E}}_{\text{EEG}} = \mathbf{E}_{\text{EEG}}^{\text{proj}} \cdot \mathbf{B} \cdot \mathbf{A} \in \mathbb{R}^{P \times D_{\text{comp}}}$. To encourage alignment between the low-rank EEG and fMRI embeddings, we optimize the MSE between the two latent embeddings as the alignment loss $\mathcal{L}_{\text{align}}$. Ablation studies on alternative objectives (e.g., cosine similarity, InfoNCE, contrastive loss) are provided in Appendix D.14, where MSE achieves the best performance.

**Training Objective.** Our framework is trained using a joint objective that combines **(1) a latent alignment loss**, $\mathcal{L}_{\text{align}}$, and **(2) a fMRI reconstruction loss**, $\mathcal{L}_{\text{recon}}$. The **reconstruction loss** is formulated as a weighted combination of MSE and spatial correlation (SCorr) to capture both absolute and relative accuracy:

$$\mathcal{L}_{\text{recon}} = \lambda_{\text{MSE}} \cdot \mathcal{L}_{\text{MSE}} + (1 - \lambda_{\text{MSE}}) \cdot \mathcal{L}_{\text{SCorr}}. \quad (3)$$

Here, the SCorr term is defined as: $\mathcal{L}_{\text{SCorr}} = 1 - \text{Corr}(\hat{Y}^{(\text{paired})}, Y^{(\text{paired})})$, where $\text{Corr}(\hat{Y}^{(\text{paired})}, Y^{(\text{paired})})$ denotes the Pearson correlation coefficient between the predicted and the ground-truth vector. This term encourages the model to match not only the absolute magnitudes but also the *relative spatial pattern* of regional activations. The overall training objective of Stage 2 is a weighted sum of the alignment and reconstruction losses:

$$\mathcal{L}_{\text{total}} = \lambda_{\text{align}} \cdot \mathcal{L}_{\text{align}} + \lambda_{\text{recon}} \cdot \mathcal{L}_{\text{recon}}, \quad (4)$$

where $\lambda_{\text{align}}$ and $\lambda_{\text{recon}}$ are hyperparameters that balance the contribution of each loss term.

## 3. Experiments

Simultaneous EEG-fMRI acquisition poses significant technical challenges, including the need for MR-compatible EEG equipment, extended setup time, and complex artifact correction procedures. As a result, publicly available simultaneous EEG-fMRI datasets remain scarce and typically small in scale. To mitigate this limitation, we leverage larger-scale unimodal fMRI pretraining to learn robust neural representations prior to cross-modal alignment. Accordingly, our experiments are designed to evaluate both model performance under limited paired supervision and generalization to unseen subjects and distributions. Specifically, our experimental evaluation includes: (i) within-distribution performance assessment on held-out resting-state EEG–fMRI data; and (ii) extensive zero-shot generalization evaluation across tasks and populations, including transfer to a task-based condition and an independent clinical EEG-only dataset from patients with Parkinson's disease. Detailed model configurations and training details are provided in Appendix C.

### 3.1. Datasets and Preprocessing

**Pretraining fMRI Dataset.** A subsample of resting-state fMRI (rs-fMRI) data from the HCP 1200-subject release (Van Essen et al., 2012) was used for pretraining. Subjects were scanned up to 4 times, twice on one day and twice on a second day. We included only those subjects who completed all four runs and were reported to have passed quality control by Xifra-Porxas et al. (2021); Power et al. (2017), resulting in 375 subjects (n = 1500 scans). These were split into 1,200 training and 300 validation scans, corresponding to approximately 720,000 training samples. Please refer to the Appendix B for preprocessing details.

**Eyes-closed Resting-state Simultaneous EEG-fMRI Datasets.** For cross-modal alignment, we utilize two eyes-closed resting-state EEG-fMRI datasets, comprising a total of 39 scans from 29 healthy subjects. **Dataset 1** is a public dataset from Li et al. (2024b). This dataset comprises 29 simultaneous EEG-fMRI scans from 22 healthy subjects, with each scan lasting 20 minutes (TR = 2.1 seconds, 32 EEG channels). **Dataset 2** is an in-house rs-EEG-fMRI dataset. It comprises 10 training scans from 7 healthy participants, with 3 individuals undergoing two scans each. Written informed consent was obtained from all participants, and all procedures were approved by the Institutional Review Board. FMRI data were acquired on a 3T Siemens Prisma scanner using a multi-echo gradient-echo EPI sequence (TR = 2.1 seconds). Simultaneous scalp EEG was recorded using a 32-channel MR-compatible system. To ensure consistency, this dataset was preprocessed using the same pipeline as Dataset 1 (See further details in Appendix B, C).

**Auditory-task Simultaneous EEG-fMRI Dataset.** To evaluate cross-task generalization, we use an auditory task dataset previously introduced in Li et al. (2024b). Further details on data collection and preprocessing can be found in Li et al. (2024b).

**Parkinson's EEG Dataset.** To evaluate real-world applicability in scenarios where fMRI is unavailable, as well as generalizability across clinical populations and age groups, we tested our model on a Parkinson's disease (PD) EEG dataset (Cavanagh, 2021) collected under resting-state and auditory oddball task conditions. The dataset comprises 64-channel EEG from 50 older adults (25 healthy controls, 25 PD patients; mean age $69.50 \pm 9.07$ years). Six subjects (3 controls and 3 patients) were excluded for the later analysis due to poor EEG quality. We refer the reader to the dataset paper (Cavanagh, 2021) for details of the EEG collection, task condition, and demographic information. For this raw dataset, we applied standard EEG preprocessing and selected the 23 EEG channels that overlap with the model's input configuration. A detailed description of the dataset and preprocessing steps is provided in Appendix B. The cleaned EEG was then segmented into 16-second windows,

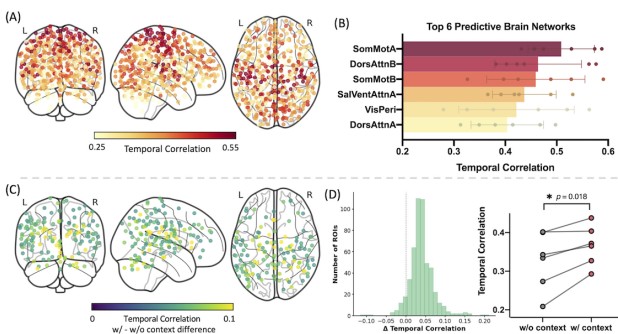

*Figure 2.* **Resting-state reconstruction performance.** (A, B) Reconstruction performance and top predictive brain networks based on network-aggregated TCorr. (C) Regional performance difference with vs. without context embedding. (D) Distribution of performance gains from context embedding across regions and samples; points denote test scans.

each of which was fed into the trained UniEFS model to extract fMRI-informed EEG latent embeddings.

**Vigilance Score.** The vigilance state is defined here as a categorical score with three classes (drowsy, intermediate, and alert) assigned to each fMRI frame. This classification is derived from EEG data based on Vigilance Algorithm Leipzig (VIGALL 2.1 add-on for Brain Vision Analyzer [1]) (Olbrich et al., 2015; Huang et al., 2015; Jawinski et al., 2019). Note that vigilance labels were shifted forward by 5 seconds (∼2 TRs) to account for the temporal delay between neural activity and the peak BOLD response. Please see Appendix B for further details.

### 3.2. Experimental Setup

**Data Preparation.** From the fMRI data in the paired and unpaired datasets, we extract ROI time courses using the DiFuMo atlas (Dadi et al., 2020) (P=512, full brain coverage). We regress out six motion-related confounds, apply a low-pass filter with a cutoff at 0.15 Hz , and z-normalize each ROI time series. The EEG data are resampled to 200 Hz and normalized by a factor of 100 to ensure most values fall within the range of -1 to 1. We select the **23** overlapped EEG channels from Dataset 1&2 for training. For each fMRI time point, a 16-second EEG window preceding the scan is extracted as model input.

**Baselines.** We compare our model with three open-source EEG-to-fMRI synthesis baselines (Kovalev et al., 2022; Li et al., 2024b;a) and state-of-the-art EEG encoders including recent foundation models (see Appendix C.2 for details). These baseline models were originally designed or benchmarked in Li et al. (2024b) under a region-specific setting. To enable comparison on multi-region reconstruction, we adapt them by modifying the final projection layer to map the latent embeddings to the entire set of ROIs, thus extend-

ing them into multi-region baselines.

### 3.3. Main Results

Our model was trained to predict held-out recordings across entire 20-minute scans. We compare UniEFS with state-of-the-art EEG-to-fMRI translation baselines (Table 1) and EEG encoders (Table 4 in Appendix D.1), finding that UniEFS consistently outperforms the others in reconstructing regional time courses and has the second-best performance in recovering FC. We refer readers to Appendix E.1 for a detailed discussion on potential factors contributing to this observation and Appendix D.15 for visualization examples of reconstruction. We further validate reconstruction quality through spectral analysis (Appendix D.7): the predicted and ground-truth power spectral densities show strong correspondence across frequencies (Pearson $r = 0.93$, $p < 0.001$), confirming that the model captures the temporal dynamics underlying fMRI fluctuations. Figure 2 presents a comprehensive evaluation of model performance, including region-wise distributions of predictive accuracy across brain areas with and without context embeddings. The results highlight the effectiveness of incorporating context embeddings, and indicate that activity in the somatomotor network is most reliably predicted from EEG, followed by the dorsal attention and salience/ventral attention networks.

To better understand the role of context information, we further analyze the effect of incorporating vigilance embeddings. Specifically, we compare model performance with and without vigilance conditioning to identify brain regions that benefit the most from this additional context. As shown in Figure 3, several regions, particularly within sensory-motor network, salience and attention-related networks, as well as one thalamus region, show marked improvement when vigilance information is included. These regions have been consistently reported in the fMRI literature as being associated with vigilance, and the spatial distribution (Figure 3(A)) closely overlaps with vigilance-related fMRI maps reported by prior studies (Liu & Falahpour, 2020; Schneider et al., 2016; Goodale et al., 2021). Under zero-shot resting-state to auditory-task generation setting (Figure 3 (D-F)), the Temporo-Parietal (TempPar) Network and Dorsal Attention Network (DAN) show the most pronounced improvements. Specifically, TempPar is known to support attentional reorienting, sensory-motor integration, and response preparation - functions that are strongly modulated by moment-to-moment arousal. In the auditory task dataset, participants must detect auditory cues and make rapid button-press responses; thus, fluctuations in alertness directly impact both auditory processing efficiency and motor readiness, processes for which TempPar plays a central role. Likewise, DAN is among the networks most sensitive to arousal and sustained attention. These results support

---

[1]https://brainvision.com/products/analyzer-2/

*Table 1.* Comparison of quantitative performance across models. MM: Whether the model is originally a multi-region model; FB: Full brain; GM: Cortical gray matter; SC: Subcortical regions; CB: Cerebellum; Conn: Full-brain functional connectivity (FC) related matrix; TCorr: Temporal correlation between prediction and ground truth; Corr: Pearson correlation between the vectorized upper-triangular entries of predicted and real FCs. **Bold**: best; underlined: second best. Paired t-test vs. our model: blue ($p < 0.05$), yellow ($p < 0.01$), red ($p < 0.001$), uncorrected.

| Model Name | MM | FB TCorr ↑ | GM TCorr ↑ | SC TCorr ↑ | CB TCorr ↑ | Conn Corr ↑ | Conn MSE ↓ |
|---|---|---|---|---|---|---|---|
| Ours | ✓ | **0.367 ± 0.052** | **0.394 ± 0.060** | **0.276 ± 0.082** | **0.247 ± 0.060** | 0.527 ± 0.084 | 0.233 ± 0.072 |
| NeuroBOLT (Li et al., 2024b) | ✗ | 0.331 ± 0.044 | 0.357 ± 0.049 | 0.258 ± 0.092 | 0.216 ± 0.046 | 0.455 ± 0.079 | 0.349 ± 0.097 |
| Li et al. (Li et al., 2024a) | ✗ | 0.312 ± 0.038 | 0.329 ± 0.037 | 0.253 ± 0.090 | 0.236 ± 0.058 | **0.535 ± 0.077** | **0.217 ± 0.065** |
| BEIRA (Kovalev et al., 2022) | ✗ | 0.171 ± 0.148 | 0.196 ± 0.170 | 0.086 ± 0.085 | 0.063 ± 0.073 | 0.459 ± 0.080 | 0.368 ± 0.090 |

the biological plausibility of our approach, demonstrating that incorporating vigilance context enables more accurate and interpretable EEG-to-fMRI translation, particularly in regions sensitive to fluctuations in arousal and attention, which is especially crucial for resting-state data.

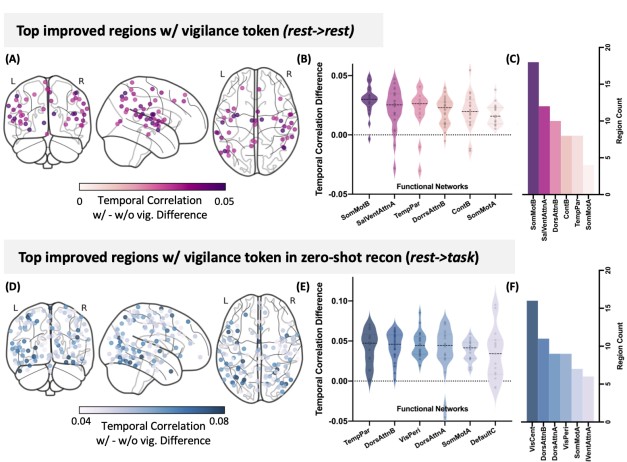

*Figure 3.* **Performance improvements with vigilance embedding.** (A-C) Within resting-state evaluation. (D-F) Zero-shot transfer from rest to task condition. (A, D) Brain regions showing the greatest performance improvement. (B, E) Distribution of region-wise improvement within brain networks. (C, F) The top 100 regions benefiting from vigilance embedding were selected and assigned to their corresponding brain networks. The 6 networks with the highest counts among these top-ranked regions are shown.

**Ablation Studies.** We conduct comprehensive ablation studies to evaluate key design choices, including loss type, training strategy, model architecture, and context embeddings in Appendix D.12– D.14. Beyond these ablations, we also analyzed the impact of mask ratio (Table 8), and patch size (Table 9) during fMRI pretraining, please see Appendix D.8 and D.9 for details.

### 3.4. Generalization and Potential Applications

In this section, we provide initial evidence that our model not only transfers effectively from resting-state to task-induced fMRI dynamics, but also yields neurophysiologically meaningful representations that generalize to entirely new populations and mitigate domain shifts through context-aware modeling. These properties highlight the potential of our approach for real-world applications where fMRI data may be unavailable or prohibitively expensive to acquire.

**Resting-state to task transfer.** We evaluate the rest-to-task transfer performance on an auditory task-based dataset Li et al. (2024b) (Figure 4). Our model generalizes well to task-induced fMRI dynamics, capturing prominent brain activity features despite not being trained on task-condition data. A comprehensive analysis of various rest-to-task transfer strategies, including fine-tuning, joint training and personalized-finetuning, is provided in Appendix D.3. To further validate the quality of the generated fMRI and its ability to reflect true subject-specific patterns in FC, we conduct the following analyses. ***First***, we evaluate *within-network time-series reconstruction* by averaging zero-shot reconstructed ROI signals within each of the 17 Yeo networks, yielding network-level fMRI time series per subject. This emphasizes larger-scale temporal dynamics beyond ROI-level variability. Quantitatively, reconstruction TCorr remains consistently strong across test samples for each network, as shown in Figure 4(F). Example reconstructions from the Salience/Ventral Attention Network (Figure 4(G)), which includes deep, non-surface regions such as the anterior insula and dorsal ACC, demonstrate that our model recovers coherent temporal structure even in networks that are typically challenging for EEG-based approaches. ***Second***, we performed a *connectome fingerprinting analysis*, as described in Appendix D.4. Notably, our model's zero-shot predicted fMRI also demonstrated high fingerprinting accuracy across full-brain, gray matter, and subcortical FC matrices (see Table 6 in Appendix). These findings suggest that the generated fMRI preserves individualized FC signatures, supporting their potential utility in downstream applications involving subject-specific brain representations (Finn et al., 2015; Mantwill et al., 2022; Lu et al., 2024).

**Context-aware modeling mitigates cross-dataset domain shift.** We systematically evaluate the effectiveness of our context-aware prompting mechanism for model generalization under a leave-one-dataset-out setting (Figure 4 (H)). We found that incorporating context prompt tokens consistently improves zero-shot performance, substantially narrowing the gap between cross- and within-dataset testing scenarios.

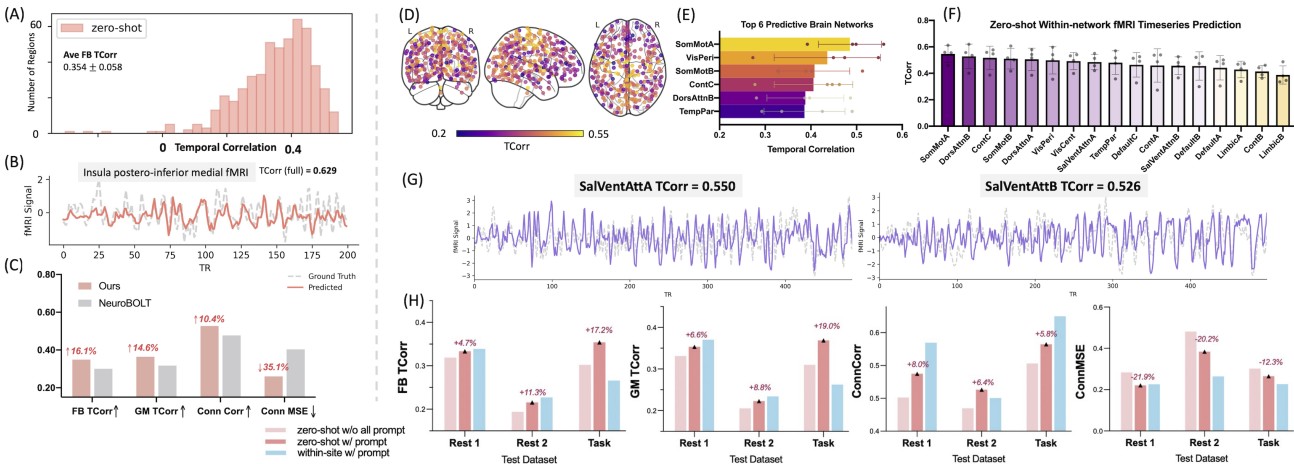

*Figure 4.* **Zero-shot cross-dataset fMRI reconstruction.** (A–G) Zero-shot rest-to-task transfer results: (A, D, E) Prediction performance distribution across all regions, with network-aggregated performance shown in (E); (B) Example reconstructed time series within the insula; (C) Comparison with the baseline; (F, G) Network-wise fMRI time series reconstruction, dots represent evaluation participants, with example visualization of reconstructed Salience/Ventral Attention Network fMRI activity shown in (G). (H) Leave-one-dataset-out performance comparing full prompt model vs. no-prompt model against within-site training.

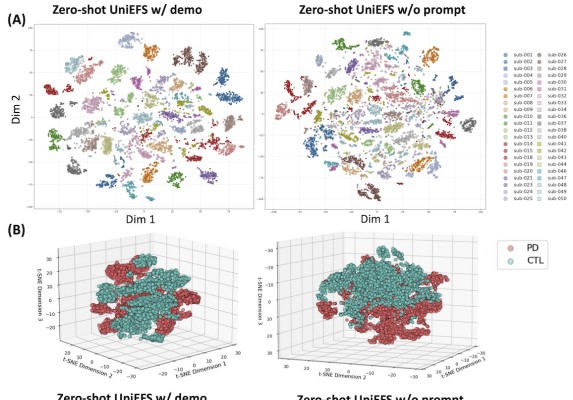

*Figure 5.* **Zero-shot EEG latent.** (A) 2D t-SNE colored by subject. (B) 3D t-SNE colored by group (PD vs. control)

Models augmented with prompt tokens not only achieve more stable performance on unseen datasets, but in several cases match or even surpass within-dataset baselines, suggesting that the learned context embeddings enable more robust generalization under distribution shifts. Detailed ablations on each prompt type are provided in Appendix D.6.

**Clinical zero-shot application with only EEG.** To evaluate generalization to clinical populations where paired fMRI data is unavailable, we tested our trained model on an external Parkinson's disease EEG dataset (Cavanagh, 2021), featuring task conditions (oddball paradigm) and demographics (elderly participants) entirely different from our training data. By directly passing EEG windows into the model without any fine-tuning, the derived EEG latent representations cleanly separate PD patients from healthy controls while maintaining strong individual-specific clustering (Figure 5). Notably, our model was trained solely to reconstruct fMRI from EEG, not to perform disease classification, yet the learned EEG-to-fMRI projection spontaneously captures clinically relevant structure. The emergence of disease-discriminative representations without explicit supervision provides preliminary evidence that fMRI-informed EEG embeddings may serve as a useful feature space for exploratory downstream analyses, including studies of disease-related neural signatures and longitudinal changes when fMRI acquisition is impractical (see details in Appendix D.5).

## 4. Discussion and Conclusion

**Contribution** We introduce **UniEFS**, a unified framework for reconstructing full-brain fMRI activity from EEG. By leveraging large-scale fMRI pretraining, domain adaptation and cross-modal alignment, our model effectively learns a mapping from EEG to whole-brain ROI-level fMRI dynamics, without relying on region-specific or subject-dependent customization. We incorporate context-aware EEG encoding using metadata-informed prompt tokens, enabling the model to account for physiological and demographic variability that modulates EEG-fMRI correspondence. Our results demonstrate that UniEFS achieves state-of-the-art performance in time-resolved fMRI signal reconstruction and shows strong potential to recover functional connectivity. These findings highlight the effectiveness of combining self-supervised fMRI representation learning with context-conditioned EEG encoding for context-aware, interpretable cross-modality translation. UniEFS paves the way for real-world applications where high-resolution, fMRI-like insights could be derived from lightweight, portable EEG systems, enabling more accessible neuroimaging in clinical, cognitive, and mobile settings.

**Limitations and Future Work** For the resting condition, our model is trained on only two paired EEG-fMRI datasets and with fewer than 32 EEG electrodes. The limited electrode coverage may impede the ability to accurately reconstruct signals from subcortical regions. Since our context-aware embeddings are designed to accommodate variability across datasets and populations, we plan to incorporate and collect additional resting-state datasets, ideally with denser EEG electrode coverage, to further enhance the model's capacity for capturing fine-grained fMRI spatial dynamics, particularly in deep brain structures.

Another limitation is that, although VIGALL and related toolboxes provide a practical approach for estimating vigilance states from EEG, the resulting vigilance estimates may be affected by dataset-specific EEG characteristics or atypical neural signatures in clinical populations. As a result, these vigilance labels may not always fully reflect the underlying vigilance state across all cohorts. This potential mismatch could affect the EEG-to-fMRI synthesis pipeline when vigilance is used as a contextual factor. However, our prompt design is modular and is not tied to a specific approach (e.g., VIGALL). The vigilance prompt could therefore be replaced or complemented by alternative physiological or neural markers of arousal, such as pupil size (Reimer et al., 2016), heart-rate variability (Xie & Ma, 2025), other EEG features, or cohort-specific vigilance annotations.

Although our results demonstrate promising EEG-to-fMRI reconstruction performance and zero-shot transfer ability, further validation with larger cohorts and more diverse task-based paradigms remains an important future direction. In particular, cognitively demanding tasks would provide a valuable opportunity to assess how well the learned EEG-to-fMRI mapping generalizes across different cognitive states and task-evoked neural dynamics. In the longer term, such advances may open avenues for clinical applications, such as noninvasive brain decoding and monitoring of cognitive or pathological states in settings where fMRI is impractical.

## Acknowledgements

The authors would like to acknowledge the Advanced MRI Section of the NINDS, NIH, where a subset of the data were collected. This work was supported in part by NIH grants R01NS112252 and F31NS143413. The authors are also grateful to Caroline Martin for assistance with data acquisition.

## Impact Statement

This paper presents work whose goal is to advance the field of machine learning for cost-effective neuroimaging, specifically by enabling the synthesis of fMRI-like brain representations from EEG signals. We believe this work has the potential for positive societal impact by democratizing access to high-resolution brain monitoring. fMRI remains largely inaccessible in under-resourced communities, outpatient settings, and for populations who cannot tolerate MRI environments (e.g., children, patients with certain implants). By enabling inference of fMRI-like information from low-cost, portable EEG systems, our approach could broaden access to advanced neuroimaging capabilities for clinical screening, cognitive assessment, and longitudinal monitoring.

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

## A. Related Work

EEG-to-fMRI reconstruction, while currently an underexplored research area, has received growing attention in recent years owing to advances in deep learning and cross-modality synthesis. Several studies (Liu & Sajda, 2023a;b; Calhas & Henriques, 2022; Lanzino et al., 2024) have proposed methods for reconstructing volume-wise fMRI spatial patterns from EEG signals. However, these approaches generally lack quantitative evaluation of temporal dynamics, that is, how brain activity evolves over time, or of temporal correlation across brain regions, which are critical readouts, underpinning analyses of functional connectivity, network dynamics, and brain state transitions. Without assessing these aspects, it remains unclear whether such models can support broader neuroscientific or clinical applications that rely on accurate reconstruction of brain-wide temporal structure. A complementary line of work has investigated fMRI time series reconstruction in specific brain regions, particularly in subcortical regions, such as early work using ridge regression (Meir-Hasson et al., 2014; Or-Borichev et al., 2023), along with more recent deep-learning studies using seq-to-seq models (Kovalev et al., 2022; Li et al., 2024a). Yet, these models are typically trained on a within-subject basis, which hinders generalizability to new individuals. Furthermore, the majority of EEG-fMRI synthesis efforts have been limited to task-based paradigms, where external cues help structure the neural responses (Kovalev et al., 2022; Li et al., 2024a; Liu & Sajda, 2023a;b; Wei et al., 2020). As a result, spontaneous resting-state activity, particularly in the natural eyes-closed condition, remains largely unexplored. To bridge this gap, recent work Li et al. (2024b) introduced a transformer-based framework for reconstructing fMRI time series in a few selected brain regions during eye-closed resting-state. While this method shows promising generalization, it still requires training separate models for each target region, limiting scalability and efficiency. A more recent work by Yao et al. (2025) demonstrated efficient cortical surface fMRI generation by conditioning a diffusion model on EEG. However, its reliance on fMRI surface maps restricts reconstruction to the cortex, leaving subcortical regions outside the model's representational space.

## B. Dataset and Preprocessing

**Pretraining fMRI Dataset: Preprocessing**  The rs-fMRI scans in HCP dataset were acquired with a temporal resolution (TR) of 0.72 seconds, a duration of 1,200 frames per run (14.4 minutes), and a spatial resolution of 2 mm isotropic. This dataset (here, used for pretraining) had been processed using the HCP Minimal Preprocessing Pipeline (Glasser et al., 2013). In addition to this standard preprocessing, we removed low-order trends (polynomials up to the 4th order) to mitigate scanner drift artifacts, and temporally downsampled the data by a factor of 2, resulting in a final temporal resolution of 1.44 seconds and 600 frames per scan. This step was performed to render the temporal resolution more comparable to conventional fMRI scans, including those of the EEG-fMRI datasets used in our study. After extracting time courses from regions of interest using DiFuMo atlas (Dadi et al., 2020), we additionally regress out six rigid-body head-motion parameters (translation and rotation), apply a low-pass filter with a cutoff at 0.15 Hz (which captures the low-frequency band typically of interest in rs-fMRI studies), and z-normalize each ROI time series.

**Details about paired EEG-fMRI Dataset 2**  Dataset 2 is a resting-state EEG-fMRI dataset collected as part of the same data acquisition effort as the auditory-task EEG-fMRI dataset previously introduced in Li et al. (2024b). It comprises 16 scans from 11 healthy participants, with 5 individuals undergoing two scans each. We used 10 scans for training, 3 scans for validation, and 3 scans for testing. Nine subjects are shared between the resting-state and auditory-task datasets, and overlapping subjects were assigned to the same train/validation/test split across both datasets, so that no subject appears in different splits across conditions. During the scans, participants rested passively with their eyes closed. Written informed consent was obtained from all participants, and all procedures were approved by the Institutional Review Board of NIH. MRI data were acquired on a 3T Siemens Prisma scanner. The T1-weighted structural images were collected with the following parameters: TR = 2200 ms, TE = 4.25 ms, flip angle = 9 deg, 1 mm isotropic. BOLD fMRI images were collected using multi-echo gradient-echo EPI sequence with TR = 2100 ms, echo times = 13.0, 29.4, and 45.7 ms, voxel size = 3 × 3 × 3 mm³, slice gap = 1 mm, matrix size = 82 × 50, 30 axial slices. MRI scanner triggers were recorded together with the EEG signals for data synchronization. The first seven volumes in fMRI data were dropped to allow magnetization to reach steady state. The fMRI preprocessing steps are kept consistent with (Li et al., 2024b). Specifically, the steps included slice-timing and motion coregistration, noise reduction using multi-echo ICA which is implemented in tedana 0.0.9a[2], alignment to an MNI152 standard template (matrix shape: 91 × 109 × 91), removal of low-order trends (up to 4th-order polynomials), and spatial smoothing (to 3mm FWHM) using AFNI [3]. Simultaneous scalp EEG was acquired using

---

[2]https://tedana.readthedocs.io/en/stable/
[3]https://afni.nimh.nih.gov/afni

a 32-channel MR-compatible system (10–20 layout, FCz reference; BrainAmps MR, Brain Products GmbH) at a sampling rate of 5 kHz. The EEG system was synchronized to the scanner's 10 MHz clock to support gradient artifact correction. Preprocessing included removal of MR-related artifacts using BrainVision Analyzer 2 (Brain Products, Munich, Germany) (Moehlman et al., 2019; Allen et al., 2000), followed by downsampling to 250 Hz. No additional filtering was applied. The full set of 32 EEG channel labels is: ['FP1', 'FP2', 'F3', 'F4', 'C3', 'C4', 'P3', 'P4', 'O1', 'O2', 'F7', 'F8', 'T7', 'T8', 'P7', 'P8', 'FZ', 'CZ', 'PZ', 'OZ', 'FC1', 'FC2', 'CP1', 'CP2', 'FC5', 'FC6', 'CP5', 'CP6', 'TP9', 'TP10', 'POZ']. For joint training across Dataset 1 (Li et al., 2024b) and Dataset 2, we used the intersection of their channel sets, resulting in 23 overlapping channels: ['FP1', 'FP2', 'F3', 'F4', 'C3', 'C4', 'P3', 'P4', 'O1', 'O2', 'F7', 'F8', 'T7', 'T8', 'P7', 'P8', 'FZ', 'CZ', 'PZ', 'OZ', 'TP9', 'TP10', 'POZ'].

**Auditory-task paired EEG-fMRI Dataset**    In Section 3.4, we evaluate whether our model, trained on resting-state data, can generalize to a different task domain without additional training. To this end, we use only the test set from the auditory-task EEG-fMRI dataset from (Li et al., 2024b). During the scans, binaural tones were presented with randomized inter-stimulus intervals (ISI), and the task included two versions differing only in tone timing: (1) a fast-ISI version (500 TRs per scan) and (2) a sparse-ISI version (693 TRs per scan). The test set comprises four scans. Two scans correspond to the fast-ISI version, and the other two to the sparse-ISI version. For additional details, please refer to (Li et al., 2024b).

**Parkinson's disease EEG Dataset**    The dataset comprises 64-channel EEG recordings (10–20 system) from 50 subjects (25 healthy controls and 25 patients with Parkinson's disease), with a mean age of $69.50 \pm 9.07$ years (control group age: $69.19 \pm 9.12$ years; patient group age: $70.07 \pm 8.79$ years). Six subjects (3 controls and 3 patients) were excluded for the later analysis due to the poor EEG quality. We refer the reader to the dataset paper (Cavanagh, 2021) for details of the EEG collection, task condition, medication and demographic information. For this raw dataset, we applied standard EEG preprocessing including band-pass filtering (0.1–75 Hz), 60 Hz notch filtering, and ICA-based artifact removal to eliminate eye-related components. EEG amplitude normalization is also performed, where the signal is divided by 100 to ensure that the majority of values fall within the range of -1 to 1. For inference, we selected the 23 EEG channels that overlap with the model's input configuration. The cleaned EEG signals were then segmented into 16-second windows with a 2-second stride (i.e., 14-second overlap), each of which was fed into the trained UniEFS model to extract fMRI-informed EEG latent embeddings.

**Vigilance States**    The vigilance state is represented as a categorical label with three classes - drowsy, intermediate, and alert - assigned to each fMRI frame. To derive these vigilance classes, we employed the automated VIGALL method (Huang et al., 2015; Jawinski et al., 2019; Olbrich et al., 2015), which classifies scalp EEG segments into five vigilance stages based on spatial power distributions. Specifically, we used the VIGALL 2.1 add-on in BrainVision Analyzer 2 to segment the preprocessed EEG into non-overlapping 1-second intervals and label each interval as one of five stages: A1, A2, A3, B1, or B2/3, corresponding to decreasing levels of alertness. Prior to staging, EEG signals were re-referenced to the average, and spherical spline interpolation was applied to reconstruct any missing channels required by the VIGALL standard. These vigilance stage labels are then grouped into 63-second epochs (corresponding to 30 fMRI time points) and the distribution of stages within each epoch is used to assign a single vigilance class, i.e., alert, intermediate, or drowsy, to that epoch, according to the following rules: (1) First, the five VIGALL stages were converted to integer values from 1 (most drowsy) to 5 (most alert); (2) The Wilcoxon signed-rank test was then applied to the integer values of each epoch to test for a significant difference of the median away from a (weighted) center value of 2.75; (3) Based on the test statistic, we assigned each epoch to one of the three vigilance classes using a z-threshold of ±1.5: epochs with significantly high or low median vigilance were labeled as alert or drowsy, respectively, while others were classified as intermediate; (4) Finally, consecutive epochs with the same vigilance label were merged to form continuous segments (Pourmotabbed et al., 2025). Note that vigilance labels were shifted forward by 5 seconds (∼2 TRs) to account for the temporal delay between neural activity and the peak BOLD response.

## C. More Implementation Details

### C.1. Hyperparameters

The default hyperparameters for the full pretraining model architecture are summarized in Table 2. The fMRI masked signal modeling (f-MSM) model is pretrained for *225 K* iterations. The checkpoint achieving the highest spatial correlation between predicted and ground-truth signals on the validation set is selected as the final pretrained model, which is further

fine-tuned for 20 epochs. The training hyperparameters of EEG-to-fMRI mapping are shown in Table 3.

*Table 2.* Hyperparameters used for stage 1: f-MSM pretraining and finetuning

| Hyperparameters | Values |
|---|---|
| batch size | pretrain: 96, fine-tune: 16 |
| learning rate | pretrain: 3e-4, fine-tune: 5.3e-5 |
| weight decay | 0.05 |
| Optimizer | AdamW |
| patch size | 1 |
| encoder embedded dim | 512 |
| mask ratio | 0.5 |
| mlp ratio | 2.0 |
| decoder embedded dim | 256 |
| encoder depth | 12 |
| encoder heads | 8 |
| decoder depth | 8 |
| decoder heads | 8 |

*Table 3.* Hyperparameters for stage 2: EEG-to-fMRI mapping

| Hyperparameters | Values |
|---|---|
| Batch size | 64 |
| Peak learning rate | 3e-4 |
| Minimal learning rate | 1e-6 |
| Learning rate scheduler | Cosine |
| Optimizer | AdamW |
| Adam $\beta$ | (0.9,0.99) |
| Weight decay | 0.05 |
| Drop path | 0.1 |
| Layer-wise learning rate decay | 0.65 |
| $\lambda_{\text{MSE}}$ | 0.5 |
| $\lambda_{\text{align}}$ | 0.8 |
| $\lambda_{\text{recon}}$ | 0.2 |

**Implementation Details** All experiments were conducted on a single NVIDIA RTX A5000 GPU using Python 3.9.12, PyTorch 2.0.0, and CUDA 11.8. The total computational cost was approximately 30 GPU hours, including 25 hours for pretraining, 1 hour for fine-tuning, and 4 hours for cross-modal alignment. We initialize EEG encoder's spatiotemporal module using pretrained weights from LaBraM-base (Jiang et al., 2024), with a token length of 200 (i.e., 1 second) and no overlap. For the multi-scale spectral module, we set the smallest scale size to $l_0 = 200$ (1 second without overlap), and use a multiscale level of 3. The training set for stage 1 consists of 1,200 scans, with 300 scans used for validation, resulting in approximately 720,000 training samples (one per time point). During stage 2, we train the model to predict fMRI signals across entire unseen scans using EEG, and use the same data partitioning strategy as in Li et al. (2024b) (an approximately 3:1:1 split for unseen-subject whole-scan reconstruction). We incorporate the training set of Dataset 2 (10 scans) as additional training data, resulting in a total of 28 training scans, 5 validation scans, and 6 test scans. Scans from the same individual are always assigned to the same split (training, validation, or test), with train/validation/test subjects strictly disjoint, since data from the same subject may have similar latent representations. For personalized finetuning in Appendix D.3, which involves within-scan splitting, we designed a 20-second gap between train/validation/test segments to mitigate temporal autocorrelation (typically ±10 seconds), ensuring no information leakage across splits. For reproducibility, a fixed random seed is used across all experiments. The functional connectivity metrics, i.e., Conn Corr and Conn MSE, are calculated using the upper triangle of the correlation matrices, as they are symmetric.

## C.2. Baselines

We compare our model against three publicly available and adaptable EEG-to-fMRI translation frameworks, all of which have been benchmarked in (Li et al., 2024b). These are the only open-source methods compatible with the datasets and experimental setup in this study and (Li et al., 2024b).

- **BEIRA** (Kovalev et al., 2022): BEIRA introduces a convolutional neural network (CNN)-based encoder-decoder architecture that translates EEG sequences into corresponding fMRI sequences in a sequence-to-sequence manner.

- **Li et al.** (Li et al., 2024a): This method extends BEIRA by incorporating an additional light-weight spectral representation learning module that leveraging sinusoidal activation function to better capture the frequency characteristics of EEG signals. It uses CNN-based downsampling and upsampling encoder-decoder blocks to perform the translation from EEG to fMRI during an eyes-open-eyes-closed task.

- **NeuroBOLT** (Li et al., 2024b): NeuroBOLT proposes a transformer-based multi-dimensional encoder for EEG-to-fMRI mapping in a seq-to-one format. It is a region-specific model, which means that models are trained separately for each region. It achieved state-of-the-art prediction performance in their resting-state dataset.

Among these baselines, the models by BEIRA and Li et al. were originally designed in a sequence-to-sequence format, where both the input and output are time series. To account for the hemodynamic delay of fMRI relative to EEG, the EEG sequence was temporally shifted by 6 seconds, i.e., the input EEG was delayed by 6 seconds to align with the corresponding fMRI response. For now we do not include CATD (Yao et al., 2025) as a baseline, since CATD operates at the surface-map level restricted to the cortex, whereas our study focuses on ROI-based reconstruction covering the whole brain, including both cortical and subcortical regions, making the two settings not directly comparable. Moreover, since the implementation of CATD has not been publicly released, accurate reproduction is not currently feasible, which would preclude a fair comparison.

We also compare our model performance with state-of-the-art EEG encoders, and results are shown in D.1.

- **CBraMod** (Wang et al., 2025a): CBraMod is a recent foundation model for EEG that follows the design of prior EEG foundation models by segmenting EEG signals into patches and pre-training via masked patch reconstruction. Building on this framework, CBraMod introduces two key innovations: (1) a criss-cross transformer backbone with parallel spatial and temporal attention mechanisms to separately capture heterogeneous dependencies in EEG, and (2) an asymmetric conditional positional encoding scheme that enables flexible adaptation to diverse EEG formats. Pre-trained on a large-scale EEG corpus, CBraMod outperforms state-of-the-art methods and demonstrates strong generalizability across up to 10 downstream BCI tasks (12 public datasets). In our experiments, we initialize the model with these pre-trained weights to provide a warm start for the EEG-to-fMRI translation task.

- **LaBraM** (Jiang et al., 2024): LaBraM (Large Brain Model) is a unified foundation model for EEG that enables cross-dataset learning by segmenting EEG signals into channel patches and using vector-quantized neural spectrum prediction to encode them into compact neural codes. Pre-trained on 2,500 hours of EEG data from 20 datasets, LaBraM achieves state-of-the-art performance in various downstream tasks such as abnormal detection, event classification, emotion recognition, and gait prediction. In our experiment, we load the pre-trained weights as initialization (version: LaBraM-base).

- **BIOT** (Yang et al., 2023): BIOT is a transformer-based foundational architecture for biomedical signal encoding. It segments EEG signals into patches and learns spatiotemporal and spectral representations from EEG, which can be applied to various downstream tasks.

- **CNNTransformer** (Peh et al., 2022): CNNTransformer is a transformer convolutional neural network originally designed for automated artifact detection in EEG.

- **STTransformer** (Song et al., 2021): STTransformer is a transformer-based spatial-temporal feature learning neural network originally designed for EEG decoding.

- **FFCL** (Li et al., 2022): FFCL is a model combining learned latent features from CNN and LSTM models for the purpose of motor imagery EEG classification.

In the NeuroBOLT experiments (Li et al., 2024b), the authors adapted all baselines to a sequence-to-one format for evaluation. Following this approach, we apply the same adaptation and further attach a shared multi-ROI MLP decoder to each EEG encoder, enabling a single model to predict the full set of ROI signals for fair comparison. Despite using pretrained EEG foundation model encoders, all baseline models are retrained from scratch to convergence under the same training protocol as ours (including optimizer, batch size, number of epochs, learning-rate schedule, and early stopping criteria). All baselines reached stable convergence under this standardized setup.

## D. Extended Experiment Results

### D.1. Prediction Performance of Other EEG Encoding Baselines

In this section, we compare the performance of our model with that of other state-of-the-art EEG encoders, which were originally developed either as general-purpose EEG foundation models (Yang et al., 2023; Jiang et al., 2024; Wang et al., 2025a) or were specifically designed for EEG decoding tasks (Song et al., 2021; Peh et al., 2022; Li et al., 2022). For a fair comparison, we attach a multi-region prediction head to each encoder to decode the full vector of fMRI ROIs. The results are reported in Table 4, where our model achieves superior performance on the majority of evaluation metrics.

*Table 4.* Full brain fMRI reconstruction: comparison with EEG encoding baselines. FB: Full brain; GM: Cortical gray matter; SC: Sub-cortical regions; CB: Cerebellum; TCorr: Temporal correlation between predicted and ground truth fMRI signals; Conn: Metric is applied on the upper triangle of the full-brain functional connectivity (FC) matrix; Conn Corr: Element-wise correlation between predicted and measured FC. **Bold**: the best; Underlined: the second best. Values are shown as mean ± std.

| Model Name | FB TCorr ↑ | GM TCorr ↑ | SC TCorr ↑ | CB TCorr ↑ | Conn Corr ↑ | Conn MSE ↓ |
|---|---|---|---|---|---|---|
| Ours | **0.367 ± 0.052** | **0.394 ± 0.060** | **0.276 ± 0.082** | 0.247 ± 0.060 | **0.527 ± 0.084** | **0.233 ± 0.072** |
| CBraMod (Wang et al., 2025a) | 0.349 ± 0.033 | 0.369 ± 0.035 | **0.276 ± 0.081** | **0.257 ± 0.059** | 0.520 ± 0.062 | 0.234 ± 0.072 |
| LaBraM-base (Jiang et al., 2024) | 0.288 ± 0.041 | 0.320 ± 0.049 | 0.207 ± 0.082 | 0.161 ± 0.037 | 0.432 ± 0.069 | 0.401 ± 0.090 |
| BIOT (Yang et al., 2023) | 0.318 ± 0.068 | 0.353 ± 0.075 | 0.211 ± 0.101 | 0.175 ± 0.053 | 0.489 ± 0.077 | 0.299 ± 0.089 |
| CNNTransformer (Peh et al., 2022) | 0.319 ± 0.085 | 0.346 ± 0.092 | 0.242 ± 0.083 | 0.197 ± 0.081 | 0.518 ± 0.078 | 0.281 ± 0.070 |
| STTransformer (Song et al., 2021) | 0.091 ± 0.062 | 0.106 ± 0.074 | 0.052 ± 0.036 | 0.048 ± 0.015 | 0.436 ± 0.075 | 0.326 ± 0.033 |
| FFCL (Li et al., 2022) | 0.298 ± 0.034 | 0.321 ± 0.030 | 0.220 ± 0.074 | 0.194 ± 0.048 | 0.471 ± 0.075 | 0.319 ± 0.085 |

### D.2. Additional Evaluation Using Log-Euclidean Distance

To further evaluate whether the predicted fMRI signals preserve functional connectivity (FC) structure, we additionally compute the Log-Euclidean distance between covariance-based FC matrices derived from the predicted and ground-truth fMRI signals. The results are reported in Table 5.

*Table 5.* Additional evaluation of covariance-based functional connectivity preservation using normalized Log-Euclidean distance.

| Model Name | Log-Euclidean Distance (Cov) ↓ |
|---|---|
| BEIRA (Kovalev et al., 2022) | 0.399 ± 0.007 |
| Li et al. (Li et al., 2024a) | **0.269 ± 0.007** |
| NeuroBOLT (Li et al., 2024b) | 0.396 ± 0.007 |
| Ours | 0.282 ± 0.011 |

### D.3. Resting-state to Task-condition Generalization

In this section, we include a more detailed evaluation of generalization from resting-state EEG-fMRI to task-based EEG-fMRI using the auditory task dataset in Li et al. (2024b). We followed the same train-test split, resulting in 9 scans for training, 3 for validation, and 4 for testing. The results are summarized below in Figure 6 and benchmarked against NeuroBOLT.

Specifically, we considered four experimental settings: **(1) Rest-to-task zero-shot generalization:** The model is trained only on resting-state data and directly evaluated on task fMRI without any further training. **(2) Fine-tuning:** The model is pretrained on resting-state data and then fine-tuned on task data. **(3) Joint training:** The model is trained on a mixture of resting-state and task data. **(4) Personalized fine-tuning:** Starting from the resting-state pretrained model, we fine-tune individually for each test scan in the task dataset using 80% of the scan for fine-tuning, 10% for validation, and 10% for testing, where we applied a 20-second gap between train/validation/test segments to mitigate temporal autocorrelation (typically ±10 seconds), ensuring no information leakage across splits. As shown in Figure 6, our model shows strong

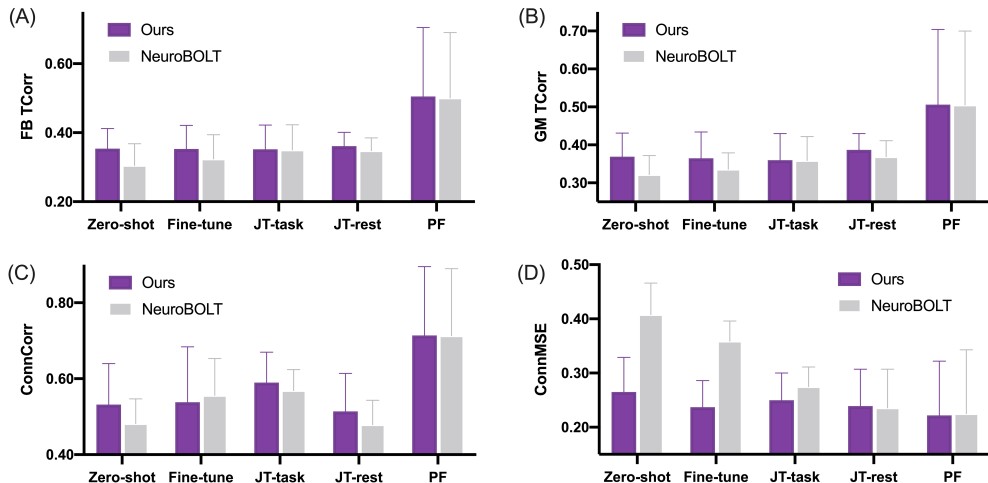

*Figure 6.* **Comparison across rest-to-task transfer and training strategies.** Performance comparison between our model and the state-of-the-art EEG-to-fMRI translation baseline, NeuroBOLT, under various evaluation setups: zero-shot transfer, fine-tuning, joint training on both rest and task scans with testing on task scans (JT-task), joint training and testing on resting-state scans (JT-rest), and personalized fine-tuning on individual task scans using a model pretrained on resting-state data (PF). (A) Full-brain temporal correlation (FB TCorr); (B) Gray matter temporal correlation (GM TCorr); (C) Spatial correlation of predicted and ground-truth functional connectivity (ConnCorr); (D) MSE of connectivity strength between real and reconstructed FC matrices (ConnMSE).

generalization from resting-state to task-based fMRI in the zero-shot setting. Fine-tuning improves performance slightly, mainly in the FC reconstruction part. Joint training helps task-fMRI FC reconstruction, but not necessarily for resting-state, which might be due to already richer variability of brain dynamics in the resting state. Although both models perform similarly in personalized fine-tuning, overall our method still performs better in most metrics especially in full-scan reconstruction scenario.

### D.4. Connectome Fingerprinting Validation of Zero-Shot FC Reconstruction

*Table 6.* Connectome fingerprinting accuracy across brain regions using ground-truth and zero-shot predicted fMRI. Our model's zero-shot outputs preserve subject-specific FC signatures. FB: full-brain; GM: gray matter; SC: subcortical regions; FC: functional connectivity; Acc: fingerprinting accuracy.

| Model | FB-FC Acc | GM-FC Acc | SC-FC Acc |
|---|---|---|---|
| Ground-truth fMRI | $100\% \pm 0\%$ | $100\% \pm 0\%$ | $100\% \pm 0\%$ |
| Zero-shot pred fMRI | $90\% \pm 10\%$ | $80\% \pm 14\%$ | $90\% \pm 10\%$ |

To further validate the quality of the zero-shot generated fMRI under task conditions as described in D.3, we performed a connectome fingerprinting analysis (Finn et al., 2015). This approach assesses whether the predicted functional connectivity (FC) patterns retain one's true brain portrait and subject-specific patterns by attempting to identify individuals based on their FC profiles.

Specifically, we selected 5 subjects from the auditory task dataset, each of whom had two scans under **different conditions (fast and sparse auditory stimulus)**. In each trial, one scan per subject was randomly assigned to a "database set", and the other scan formed the "target set". For each target FC matrix, we computed Pearson correlations with all database matrices (using vectorized upper-triangular edge values), and predicted subject identity by selecting the database matrix with the highest similarity. This procedure was repeated across all 16 possible permutations (trials), and the average identification accuracy was reported.

As shown in Table 6, ground-truth fMRI achieved perfect identification accuracy. Notably, our model's predicted fMRI also demonstrated high fingerprinting accuracy across full-brain, gray matter, and subcortical FC matrices. These findings suggest that the generated fMRI signals preserve individualized functional connectivity signatures, supporting their potential utility in downstream applications involving subject-specific brain representations (Finn et al., 2015; Mantwill et al., 2022;

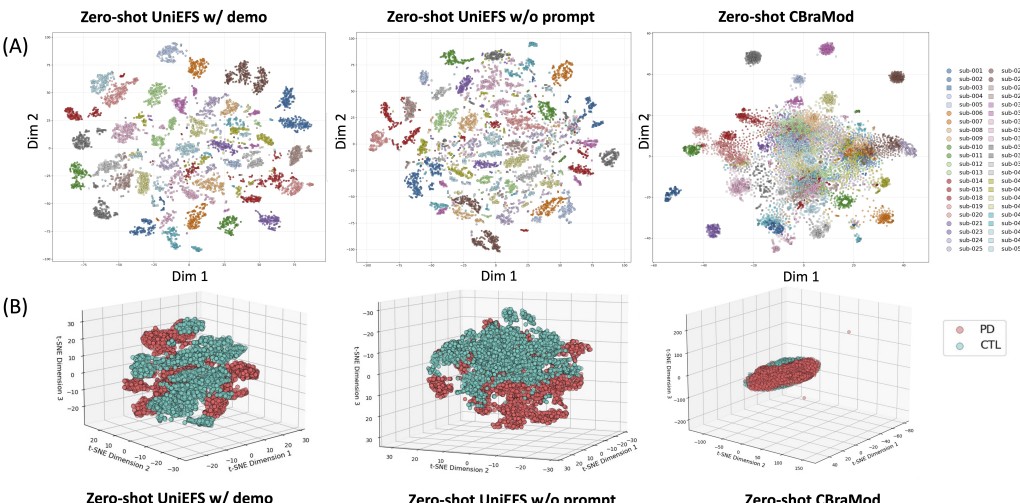

*Figure 7.* **Zero-shot EEG latent generated by UniEFS and CBraMod**. (A) 2-D t-SNE with each subject visualized with different colors. (B) 3-D t-SNE with colors distinguishing Parkinson patient (PD) and control (CTL) groups

Lu et al., 2024). Further tests using scans collected from the same individuals across different sessions/days would be important for validating these initial fingerprinting results.

### D.5. Zero-shot Evaluation under EEG-only Scenarios

To evaluate the model under real-world EEG-only scenarios where paired fMRI is unavailable, we considered two publicly available clinical EEG datasets: a Parkinson's disease (PD) dataset collected during an oddball task (Cavanagh, 2021), and the EEG subset of BrainLat, a multi-site multimodal cohort involving Alzheimer's disease (AD) and healthy control participants (Prado et al., 2023). These datasets differ substantially from our training data in terms of participant population, acquisition setting, task condition, and EEG channel configuration. Since the AD and control groups in BrainLat exhibited highly separable representations across multiple models (see Table 7), we focus our main exploration on the more challenging PD dataset.

**Parkinson's disease EEG dataset** To evaluate real-world applicability where fMRI is unavailable, along with generalizability to a clinical population, we tested our model on a Parkinson's EEG dataset (Cavanagh, 2021) with task conditions (oddball task) and demographic distribution (elderly participants) entirely different from our training data. We refer the reader to the dataset paper for details of the EEG collection, task condition, and demographic information. For this raw dataset, we performed standard EEG preprocessing, including band-pass filtering (0.1–75 Hz), 60 Hz notch filtering, and ICA-based artifact removal to eliminate eye-related components. The cleaned EEG was then epoched into 16-second windows with a 2-second stride (i.e., 14-second overlap). We fed each EEG window into our trained UniEFS model to extract the fMRI-informed EEG latent embedding. We then compared the t-SNE plots of our model's embeddings versus the embeddings from the state-of-the-art EEG foundation model CBraMod (Wang et al., 2025a) as shown in Figure 7. Note that we did not use the VIGALL-derived vigilance prompt for this EEG-only clinical evaluation. Since the applicability of VIGALL to clinical populations with atypical neural signatures remains to be further validated, we conservatively omitted the vigilance prompt in this Parkinson's disease evaluation.

To quantitatively assess representation quality, we performed subject-level linear probing on the extracted latent embeddings. Specifically, we trained a logistic regression classifier using 5-fold cross-validation for binary classification of Parkinson's disease patients versus healthy controls. This analysis is intended to evaluate whether the zero-shot fMRI-informed latent space contains clinically relevant structure, rather than to establish a diagnostic model. The results are summarized in Table 7.

In summary, we have following key observations:

(1) **Disease-related organization in the zero-shot latent space:** The t-SNE visualization shows a visible separation trend between Parkinson's disease patients and healthy controls, without any fine-tuning on this dataset. This suggests that the

zero-shot fMRI-informed EEG representation may capture disease-related neural structure that transfers to an unseen clinical population. Given the limited sample size, we interpret this result as preliminary evidence of representation quality rather than definitive disease classification performance.

(2) **Quantitative support from linear probing:** The subject-level linear probing results in Table 7 provide quantitative support for this observation. UniEFS with context tokens achieves the strongest overall PD classification performance, even though our model is trained on only 28 scans rather than a large-scale EEG corpus like existing EEG foundation models. Removing context tokens or fMRI pretraining reduces performance. This suggests that both contextual information and fMRI-informed pretraining contribute to representation quality under this zero-shot clinical setting.

(3) **Subject-specific structure in the learned representation:** Samples from the same subject tend to cluster together in the latent space, suggesting that the learned representation also preserves stable individual-specific EEG patterns even without explicit subject labels. This observation is consistent with our connectome fingerprinting results, where subject-specific information is retained in the predicted fMRI-derived representations.

Together, these results suggest that UniEFS learns transferable EEG representations with both disease-related and subject-specific structure, supporting its potential utility as a representation learning backbone for future EEG-only clinical studies.

*Table 7.* Subject-level linear probing evaluation for PD-versus-HC and AD-versus-HC classification on external EEG-only clinical datasets. Logistic regression with 5-fold cross-validation is used to evaluate the quality of zero-shot latent representations. PD: Parkinson's disease; AD: Alzheimer's disease; HC: healthy controls.

| Model | PD Acc ↑ | PD F1 ↑ | PD AUC ↑ | AD AUC ↑ |
|---|---|---|---|---|
| UniEFS (w/ context) | **0.75** | **0.73** | **0.87** | **1.00** |
| UniEFS (w/o context) | 0.68 | 0.67 | 0.79 | **1.00** |
| UniEFS (w/o pretrain) | 0.60 | 0.62 | 0.70 | 0.91 |
| LaBraM (Jiang et al., 2024) | 0.73 | 0.71 | 0.86 | **1.00** |
| CBraMod (Wang et al., 2025a) | 0.70 | 0.67 | 0.71 | **1.00** |

## D.6. Leave-one-dataset-out Test

Figure 8 presents a detailed comparison between leave-one-dataset-out generalization and within-dataset training with three datasets (Rest Dataset 1, Rest Dataset 2, and Auditory-task Dataset), along with a comprehensive ablation of the different prompt components. Removing individual prompts (dataset tokens, vigilance, demographic cues) leads to consistent drops in zero-shot performance, highlighting that each component contributes complementary contextual information.

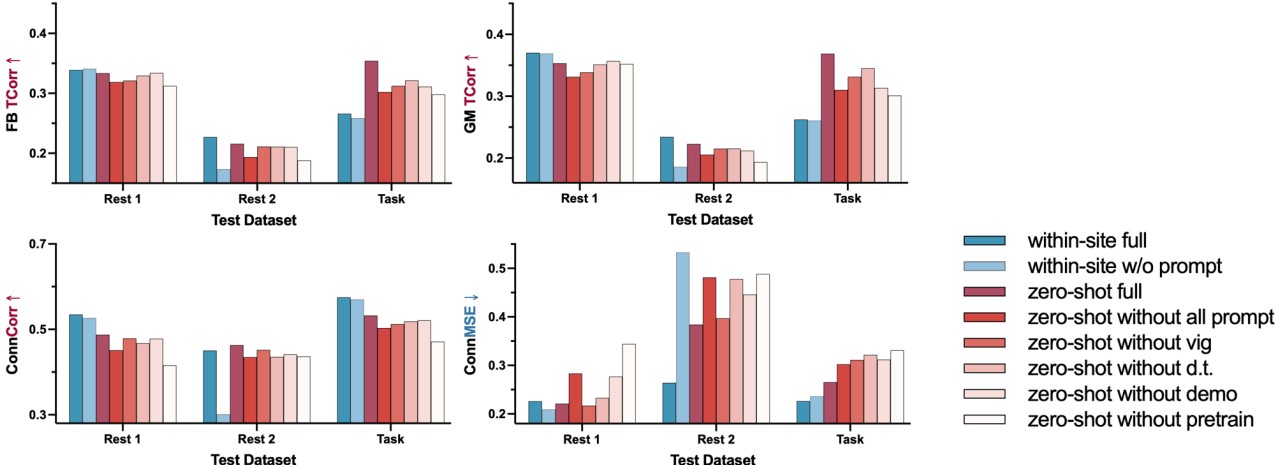

*Figure 8.* **Evaluation of leave-one-dataset-out vs. within-dataset performance under different prompt ablation settings.** Comprehensive ablation analysis of each prompt component, including dataset tokens (d.t.), vigilance (vig), demographic prompts (demo), and pretraining, evaluated under both cross- and within-dataset settings. Across test datasets (Rest 1, Rest 2, and Task), prompts consistently improve generalization, and removing specific prompt types or pretraining reveals their individual contributions to mitigating domain shifts.

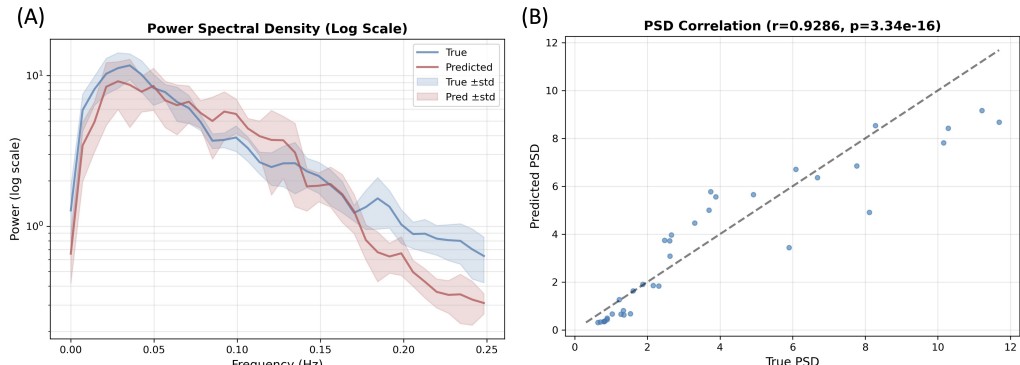

*Figure 9.* **Real and generated fMRI spectrum comparison**

### D.7. Physiological Plausibility Check

To further evaluate the physiological plausibility of the generated fMRI signal, we calculated and compared the power spectrum of real and generated fMRI signals across all regions. As shown in Figure 9, the predicted BOLD time series recovers the characteristic low-frequency PSD profile of real fMRI, including the dominant $< 0.1$ Hz band and the expected log-linear decay. Moreover, the predicted and true PSDs exhibit very high correspondence across frequencies (Pearson r = 0.93, $p < 0.001$; Figure 9 (B)), demonstrating that the model captures the correct temporal dynamics underlying fMRI fluctuations. This analysis complements our TCorr and FC evaluations by confirming that the reconstructed signals preserve known neurophysiological properties of the BOLD response.

### D.8. Effect of Masking Ratio during Pretraining

Table 8 shows the impact of different mask ratios on performance. We found that a mask ratio of 0.5 yielded the best results. In typical Masked Autoencoder (MAE) training, a high mask ratio (e.g., 0.75) is often chosen to challenge the model to recover missing information and learn robust representations (He et al., 2022; Chen et al., 2023). However, in our case, a mask ratio of 0.75 did not yield the best performance for fMRI ROI data. The suboptimal performance of this high mask ratio may be due to the significant loss of inter-ROI correlation information, which is necessary for recovering full brain fMRI patterns. In other words, since we have already averaged the (voxel-wise) signals within regions to obtain the ROI data, much of the redundancy in voxel-wise signals has been reduced. Masking too much information hampers the model's ability to capture the relationships between ROIs, which are essential for meaningful EEG-to-fMRI mapping. Preserving spatial continuity and functional connectivity is critical for the model to learn accurate representations. While when the mask ratio is small, it makes the task too easy for the model to learn complex patterns and may overfit to the existing information, leading to suboptimal generalization and performance on unseen data.

*Table 8.* Influence of mask ratio in f-MSM

| Mask ratio | FB TCorr ↑ | GM TCorr ↑ | SC TCorr ↑ | CB TCorr ↑ | Conn Corr ↑ | Conn MSE ↓ |
|---|---|---|---|---|---|---|
| 0.25 | $0.352 \pm 0.044$ | $0.378 \pm 0.048$ | $0.272 \pm 0.080$ | $0.234 \pm 0.052$ | $0.511 \pm 0.078$ | $0.274 \pm 0.086$ |
| **0.50** | $\mathbf{0.367 \pm 0.052}$ | $\mathbf{0.394 \pm 0.060}$ | $\mathbf{0.276 \pm 0.082}$ | $\mathbf{0.247 \pm 0.060}$ | $\mathbf{0.527 \pm 0.084}$ | $\mathbf{0.233 \pm 0.072}$ |
| 0.75 | $\underline{0.357 \pm 0.058}$ | $\underline{0.385 \pm 0.066}$ | $0.269 \pm 0.090$ | $\underline{0.241 \pm 0.061}$ | $\underline{0.518 \pm 0.086}$ | $\underline{0.263 \pm 0.085}$ |

### D.9. Impact of Patch Size

*Table 9.* Influence of patch size in f-MSM

| Patch size | FB TCorr ↑ | GM TCorr ↑ | SC TCorr ↑ | CB TCorr ↑ | Conn Corr ↑ | Conn MSE ↓ |
|---|---|---|---|---|---|---|
| **1** | $\mathbf{0.367 \pm 0.052}$ | $\mathbf{0.394 \pm 0.060}$ | $0.276 \pm 0.082$ | $\mathbf{0.247 \pm 0.060}$ | $\mathbf{0.527 \pm 0.084}$ | $\mathbf{0.233 \pm 0.072}$ |
| 2 | $0.347 \pm 0.069$ | $0.375 \pm 0.079$ | $0.256 \pm 0.088$ | $0.222 \pm 0.057$ | $0.475 \pm 0.083$ | $0.322 \pm 0.088$ |
| 4 | $0.355 \pm 0.040$ | $0.378 \pm 0.045$ | $\mathbf{0.281 \pm 0.078}$ | $\underline{0.245 \pm 0.054}$ | $0.491 \pm 0.093$ | $0.282 \pm 0.081$ |
| 8 | $\underline{0.362 \pm 0.062}$ | $\underline{0.387 \pm 0.070}$ | $\mathbf{0.281 \pm 0.087}$ | $\underline{0.245 \pm 0.063}$ | $\underline{0.500 \pm 0.082}$ | $\underline{0.257 \pm 0.072}$ |

In our default setting, the model uses a patch size of 1, where each token corresponds to a single brain region. This approach

is grounded in the understanding that, unlike images - where adjacent pixels often share semantic content due to spatial continuity (Dosovitskiy et al., 2020) - the ordering of regions of interest (ROIs) in a brain data vector does not inherently reflect anatomical proximity or functional similarity. Consequently, neighboring entries in the ROI vector may correspond to brain areas that are neither anatomically adjacent nor functionally related. By representing each ROI as a separate token, the model avoids imposing artificial spatial assumptions and allows for the learning of functional relationships based on actual connectivity patterns rather than from an arbitrary ordering. Here we compare the performance across different patch sizes in transformer of f-MSM. For patch sizes larger than 1, the patched data are transformed into embeddings using a 1D convolutional layer with the stride equal to the patch size. As shown in Table 9, a patch size of 1 achieves the best performance among most of the metrics compared with the larger patch sizes.

### D.10. Impact of EEG Input Length

*Table 10.* Performance under different EEG input lengths.

| EEG Length | FB TCorr ↑ | GM TCorr ↑ | SC TCorr ↑ | CB TCorr ↑ | ConnCorr ↑ | ConnMSE ↓ |
|---|---|---|---|---|---|---|
| 16 s | $0.367 \pm 0.052$ | $0.394 \pm 0.060$ | $0.276 \pm 0.082$ | $0.247 \pm 0.060$ | $0.527 \pm 0.084$ | $0.233 \pm 0.072$ |
| 12 s | $0.344 \pm 0.049$ | $0.368 \pm 0.057$ | $0.264 \pm 0.069$ | $0.235 \pm 0.054$ | $0.548 \pm 0.065$ | $0.187 \pm 0.058$ |
| 8 s | $0.303 \pm 0.059$ | $0.330 \pm 0.067$ | $0.225 \pm 0.055$ | $0.211 \pm 0.047$ | $0.519 \pm 0.074$ | $0.246 \pm 0.077$ |
| 4 s | $0.157 \pm 0.030$ | $0.157 \pm 0.044$ | $0.180 \pm 0.039$ | $0.166 \pm 0.043$ | $0.314 \pm 0.040$ | $0.249 \pm 0.057$ |

### D.11. Performance with Different DiFuMo Atlas Granularities

Table 11 reports EEG-to-fMRI reconstruction performance across different DiFuMo atlas granularities (256, 512, and 1024 ROIs). Overall, the 256-ROI atlas yields the highest reconstruction accuracy across all metrics, followed by 512 ROIs, while 1024 ROIs shows the largest performance drop. This trend reflects the increasing difficulty of predicting finer-grained, higher-dimensional fMRI representations from limited EEG information: as ROI granularity increases, each region becomes smaller and noisier, making both temporal and connectivity reconstruction more challenging. These results suggest that moderate atlas resolutions (e.g., 256-512) strike a favorable balance between spatial detail and predictable signal quality.

*Table 11.* Performance with different DiFuMo atlas granularities.

| ROI | FB TCorr ↑ | GM TCorr ↑ | SC TCorr ↑ | CB TCorr ↑ | ConnCorr ↑ | ConnMSE ↓ |
|---|---|---|---|---|---|---|
| 256 | $0.387 \pm 0.059$ | $0.415 \pm 0.064$ | $0.325 \pm 0.084$ | $0.260 \pm 0.074$ | $0.591 \pm 0.076$ | $0.213 \pm 0.088$ |
| 512 | $0.367 \pm 0.052$ | $0.396 \pm 0.058$ | $0.251 \pm 0.037$ | $0.247 \pm 0.060$ | $0.527 \pm 0.084$ | $0.233 \pm 0.072$ |
| 1024 | $0.321 \pm 0.036$ | $0.358 \pm 0.043$ | $0.262 \pm 0.072$ | $0.204 \pm 0.058$ | $0.485 \pm 0.059$ | $0.304 \pm 0.072$ |

### D.12. Ablation of EEG Encoder Modules

Our EEG encoder follows the NeuroBOLT design, which combines a temporal–spatial representation module with a multi-scale spectral representation module. The temporal–spatial module models EEG dynamics across electrode channels and time, while the spectral module extracts frequency-domain representations from EEG spectrograms computed at multiple temporal scales. In particular, using multiple STFT window sizes allows the encoder to capture complementary temporal–frequency information: shorter windows provide finer temporal resolution, whereas longer windows provide finer frequency resolution. The outputs of these two modules are then fused to provide a joint EEG representation for fMRI reconstruction. We refer readers to the original NeuroBOLT paper (Li et al., 2024b) for full architectural details.

Table 12 summarizes the ablation results on the EEG encoder. These results show that the temporal–spatial module and the multi-scale spectral module contribute complementary information for EEG-to-fMRI reconstruction. The performance drop after removing either component suggests that the final encoder design is not arbitrary, but is supported by both the original NeuroBOLT analysis and our task-specific ablation.

### D.13. Ablation on Training Strategy and Context Tokens

We conduct ablation studies to evaluate the contribution of (1) our two-stage training strategy and (2) the context prompt tokens. Results are summarized in Table 13.

**Training Strategy.** We first compare our full model against variants with different training configurations. Removing pretraining leads to a significant performance drop, particularly in connectivity metrics, indicating that the pretrained fMRI

*Table 12.* Ablation study on architectural components in EEG encoder. Mean $\pm$ std. Colors denote significance vs. full model derived by paired t-test: red ($p < 0.001$), yellow ($p < 0.01$), blue ($p < 0.05$). Bold indicates best performance per column.

| Model Variant | FB Corr ↑ | GM TCorr ↑ | SC TCorr ↑ | CB TCorr ↑ | ConnCorr ↑ | ConnMSE ↓ |
|---|---|---|---|---|---|---|
| Ours (Full) | **0.367 $\pm$ 0.052** | **0.396 $\pm$ 0.058** | 0.251 $\pm$ 0.037 | **0.247 $\pm$ 0.060** | 0.527 $\pm$ 0.084 | 0.233 $\pm$ 0.072 |
| Ours (no MSS) | 0.189 $\pm$ 0.076 | 0.206 $\pm$ 0.092 | 0.126 $\pm$ 0.042 | 0.139 $\pm$ 0.074 | 0.122 $\pm$ 0.032 | 0.407 $\pm$ 0.098 |
| Ours (no TS) | 0.350 $\pm$ 0.065 | 0.376 $\pm$ 0.075 | **0.262 $\pm$ 0.083** | 0.226 $\pm$ 0.055 | **0.530 $\pm$ 0.074** | **0.230 $\pm$ 0.080** |

encoder provides essential priors for capturing inter-regional dependencies. Skipping fine-tuning (i.e., using the pretrained decoder directly) yields comparable spatial correlation but slightly worse connectivity reconstruction.

**Context Tokens.** We further examine the contribution of each context token. We found that removing any of the context tokens resulted in a drop in performance, while removing all together leads to a more substantial drop across all metrics. The number of dataset tokens serves as a tunable hyperparameter. We found that using five tokens was sufficient to effectively handle the two training datasets used in our experiments.

*Table 13.* Ablation study on training strategy and context tokens. vig.: vigilance token; demo.: demographic token; d.t.: dataset token; w/o: without. Paired t-test significance between our model and each ablation is shown using color codes: blue ($p < 0.05$), yellow ($p < 0.01$), red ($p < 0.001$), uncorrected.

| Model Type | FB TCorr | GM TCorr | Conn Corr | Conn MSE |
|---|---|---|---|---|
| Full (context + 5 d.t.) | **0.367 $\pm$ 0.052** | **0.394 $\pm$ 0.060** | 0.527 $\pm$ 0.084 | **0.233 $\pm$ 0.072** |
| w/o fine-tune | 0.365 $\pm$ 0.056 | 0.391 $\pm$ 0.064 | 0.527 $\pm$ 0.066 | 0.267 $\pm$ 0.075 |
| w/o pretrain | 0.329 $\pm$ 0.033 | 0.361 $\pm$ 0.039 | 0.315 $\pm$ 0.074 | 0.439 $\pm$ 0.102 |
| w/o vig. | 0.357 $\pm$ 0.040 | 0.381 $\pm$ 0.044 | **0.532 $\pm$ 0.076** | 0.237 $\pm$ 0.070 |
| w/o demo. | 0.351 $\pm$ 0.060 | 0.378 $\pm$ 0.068 | 0.503 $\pm$ 0.081 | 0.271 $\pm$ 0.082 |
| w/o demo. & vig. | 0.327 $\pm$ 0.075 | 0.356 $\pm$ 0.089 | 0.491 $\pm$ 0.079 | 0.288 $\pm$ 0.089 |
| w/o age | 0.351 $\pm$ 0.047 | 0.376 $\pm$ 0.052 | 0.513 $\pm$ 0.088 | 0.261 $\pm$ 0.078 |
| w/o sex | 0.358 $\pm$ 0.041 | 0.382 $\pm$ 0.044 | 0.529 $\pm$ 0.079 | 0.235 $\pm$ 0.071 |
| w/o d.t. | 0.355 $\pm$ 0.049 | 0.380 $\pm$ 0.055 | 0.527 $\pm$ 0.081 | 0.234 $\pm$ 0.069 |
| 1 d.t. | 0.349 $\pm$ 0.051 | 0.375 $\pm$ 0.059 | 0.494 $\pm$ 0.084 | 0.288 $\pm$ 0.085 |
| 10 d.t. | 0.356 $\pm$ 0.048 | 0.381 $\pm$ 0.054 | 0.527 $\pm$ 0.081 | 0.236 $\pm$ 0.070 |

## D.14. Ablation of Loss Functions

**Loss Component Ablation** We further analyze the contribution of each term in the combined loss by performing a loss component ablation. Starting from the full model, we remove one loss term at a time while keeping all other components and training settings fixed. Table 14 summarizes the results. Removing the alignment loss $\mathcal{L}_{\text{align}}$ leads to a consistent degradation across temporal correlation and connectivity metrics, indicating its importance for cross-modal representation alignment. Eliminating the reconstruction loss $\mathcal{L}_{\text{recon}}$ results in a substantial performance drop on all metrics, highlighting its critical role in preserving signal fidelity. In contrast, removing the spatial correlation loss $\mathcal{L}_{\text{SCorr}}$ or the MSE-based connectivity loss $\mathcal{L}_{\text{MSE}}$ causes more moderate but still statistically significant degradations, particularly in functional connectivity metrics. Overall, these results demonstrate that each loss component contributes to the final performance, with the reconstruction and alignment terms playing a particularly central role in the proposed framework.

*Table 14.* Ablation study on loss components. w/o: without. Paired t-test significance between our model and each ablation is shown using color codes: blue ($p < 0.05$), yellow ($p < 0.01$), red ($p < 0.001$), uncorrected.

| Model Type | FB TCorr | GM TCorr | Conn Corr | Conn MSE |
|---|---|---|---|---|
| Full | **0.367 $\pm$ 0.052** | **0.394 $\pm$ 0.060** | **0.527 $\pm$ 0.084** | **0.233 $\pm$ 0.072** |
| w/o $\mathcal{L}_{\text{align}}$ | 0.339 $\pm$ 0.052 | 0.367 $\pm$ 0.055 | 0.502 $\pm$ 0.082 | 0.280 $\pm$ 0.083 |
| w/o $\mathcal{L}_{\text{recon}}$ | 0.147 $\pm$ 0.022 | 0.150 $\pm$ 0.025 | 0.172 $\pm$ 0.052 | 0.266 $\pm$ 0.042 |
| w/o $\mathcal{L}_{\text{SCorr}}$ | 0.355 $\pm$ 0.049 | 0.384 $\pm$ 0.056 | 0.500 $\pm$ 0.083 | 0.293 $\pm$ 0.088 |
| w/o $\mathcal{L}_{\text{MSE}}$ | 0.347 $\pm$ 0.050 | 0.376 $\pm$ 0.058 | 0.481 $\pm$ 0.085 | 0.317 $\pm$ 0.092 |

**Alignment Loss Ablation** To study the impact of different alignment objectives, we compare several commonly used loss functions for cross-modal representation alignment during Stage 2, including mean squared error (MSE), InfoNCE, contrastive loss, cosine similarity, and their combinations. All variants are evaluated under the same training protocol with identical model architecture and hyperparameters. The results are summarized in Table 15. Overall, MSE yields the

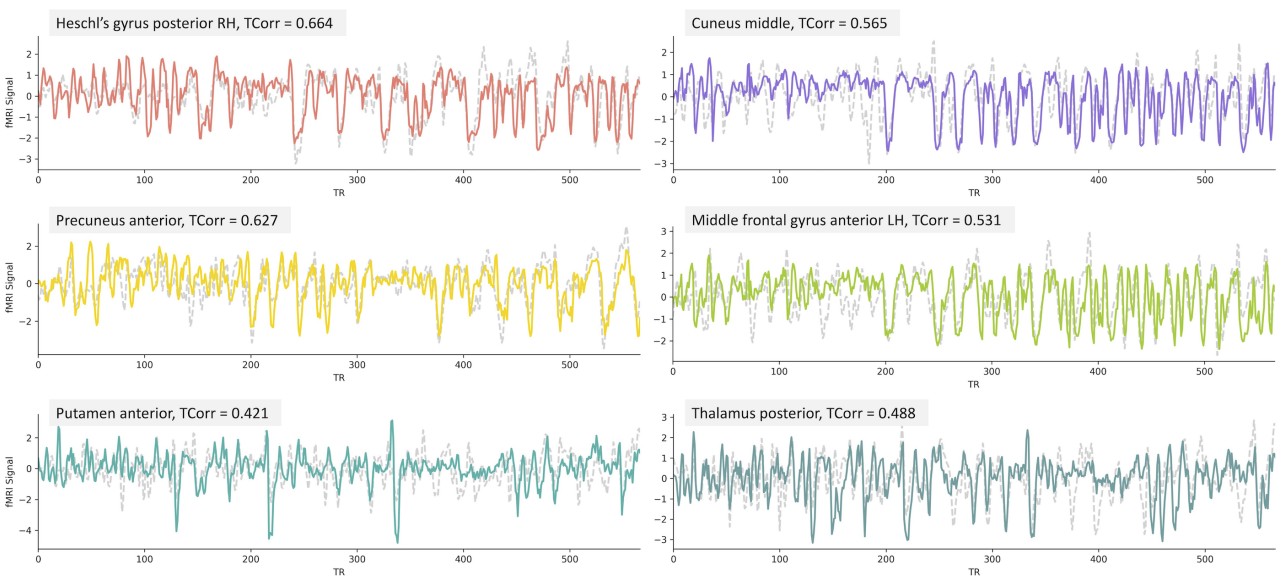

*Figure 10.* Held-out whole-scan reconstruction examples in resting-state.

most stable and consistently strong performance across temporal correlation metrics, achieving the highest or near-highest scores on FB TCorr, GM TCorr, and CB TCorr. While alternative objectives such as InfoNCE and cosine similarity exhibit competitive performance on certain connectivity-based metrics (e.g., ConnCorr and ConnMSE), they introduce substantially larger variance across subjects, particularly in subcortical regions (SC TCorr), leading to less reliable behavior. This instability is also reflected in the high standard deviations observed for several contrastive-based objectives. As a result, we adopt MSE as the alignment loss in our final model.

*Table 15.* Experiment on different alignment losses.

| Loss Type | FB TCorr ↑ | GM TCorr ↑ | SC TCorr ↑ | CB TCorr ↑ | ConnCorr ↑ | ConnMSE ↓ |
|---|---|---|---|---|---|---|
| MSE (Ours) | **0.367 ± 0.052** | **0.396 ± 0.058** | 0.251 ± 0.0371 | **0.247 ± 0.060** | 0.527 ± 0.084 | 0.233 ± 0.072 |
| InfoNCE | 0.346 ± 0.028 | 0.372 ± 0.017 | **0.270 ± 0.270** | 0.217 ± 0.074 | 0.537 ± 0.167 | **0.180 ± 0.001** |
| Contrastive | 0.329 ± 0.005 | 0.354 ± 0.007 | 0.256 ± 0.441 | 0.207 ± 0.018 | 0.516 ± 0.150 | 0.204 ± 0.002 |
| Cosine | 0.362 ± 0.054 | 0.391 ± 0.063 | 0.266 ± 0.074 | 0.232 ± 0.053 | **0.558 ± 0.091** | 0.212 ± 0.071 |
| MSE+Cosine | 0.352 ± 0.050 | 0.379 ± 0.076 | 0.256 ± 0.435 | 0.236 ± 0.039 | 0.511 ± 0.153 | 0.246 ± 0.070 |

### D.15. Unseen Resting-state Scan Reconstruction Examples

In this section, we present several examples of held-out resting-state fMRI scan reconstructions (Figure 10). These examples demonstrate the ability of our unified model to efficiently reconstruct entire resting-state fMRI scans across a wide time range using a single model.

## E. Additional Discussion

### E.1. Discussion on FC Recovery and Comparison with Baselines

While comparing our model with other EEG-to-fMRI translation baselines, we observe that UniEFS achieves the second-best performance in recovering functional connectivity (FC). One plausible reason our model does not outperform the CNN-based approach by Li et al. (2024a) in FC reconstruction is that FC is computed using Pearson correlation, which is highly sensitive to noise. Even minor prediction deviations can result in amplified discrepancies in pairwise correlations. The CNN baseline tends to produce smoother and more regularized outputs, which may suppress high-frequency fluctuations and thus yield more stable FC metrics—particularly in small-scale evaluation settings. In contrast, our model prioritizes frame-wise fidelity and regional dynamics, which may introduce local variability despite capturing more detailed temporal patterns. Notably, despite this, our model offers overall more consistent and strong performance across diverse evaluation settings.

## E.2. Discussion on the Complementary Roles of Reconstruction and Alignment Loss

From Table 14, we observe that during the alignment stage the reconstruction loss contributes the largest performance gain, while the improvement from the alignment loss is numerically smaller. Importantly, however, the gains from the alignment term are statistically significant and consistently positive across all subjects, indicating that its effect is systematic rather than noise-driven. This behavior aligns with the distinct roles of the two losses in our architecture. In stage-2, the fMRI decoder is frozen; thus, the reconstruction loss provides the primary source of task-specific semantic supervision, guiding the EEG encoder to produce latent representations that remain decodable through the fixed decoder. By contrast, the alignment loss encourages geometric proximity between EEG and fMRI latents but does not account for the decoder's nonlinear inversion geometry. Because the pretrained fMRI latent space forms a curved and structured manifold, even small off-manifold deviations - while close in Euclidean distance - may decode into semantically incorrect fMRI patterns. Consequently, the reconstruction loss naturally leads to larger numerical improvements, whereas the alignment loss serves as a regularizer that improves latent-space geometry, stability, and cross-subject consistency. This explains why its contribution is more modest in magnitude yet remains statistically reliable.

## E.3. Clarification on Reconstruction vs. Future-Frame Prediction

The model reconstructs the fMRI signal associated with the neural activity expressed in the preceding EEG window. This choice follows the physiology of the hemodynamic response, where the BOLD signal at time t primarily reflects neural events that occurred several seconds earlier. Accordingly, our formulation focuses on recovering the temporally aligned BOLD representation rather than predicting future fMRI states beyond what is supported by the EEG window. This differs from multi-step forecasting approaches, which aim to predict future frames and require additional considerations such as modeling fMRI autocorrelation and controlling temporal information leakage. Such forecasting extensions are conceptually distinct and would be an interesting future direction to explore.

